

**Coastal Sources, Sinks and Strong Organic Complexation**
**of Dissolved Cobalt within the US North Atlantic**
**GEOTRACES Transect GA03**
**Abigail. E. Noble[1*], Daniel. C. Ohnemus[1#], Nicholas J. Hawco[1], Phoebe J. Lam[1†],**
**and Mak A. Saito[1]**
[1] Woods Hole Oceanographic Institution, Woods Hole, MA, USA
[*] now at: Gradient, 20 University Road, Cambridge, MA, USA
[#] now at: Bigelow Laboratory for Ocean Sciences, East Boothbay, ME, USA
[†] now at: University of California Santa Cruz, Santa Cruz, CA, USA
Correspondence to: M. A. Saito (msaito@whoi.edu)




**Abstract**
Cobalt is the scarcest of metallic micronutrients and displays a complex biogeochemical
cycle. This study examines the distribution, chemical speciation, and biogeochemistry of
dissolved cobalt during the U.S. North Atlantic GEOTRACES Transect expeditions
(GA03/3_e), which took place in the fall of 2010 and 2011. Two major subsurface sources of
cobalt to the North Atlantic were identified by increased abundances of dissolved cobalt
relative to surrounding waters. The more prominent of the two was a large plume of cobalt
emanating from the African coast off the Eastern Tropical North Atlantic coincident with the
oxygen minimum zone (OMZ) likely due to a confluence of processes including reductive
dissolution, biouptake and remineralization, and aeolian dust deposition. This occurrence of
this plume in an OMZ with oxygen above suboxic levels implies a high threshold for
persistence of dissolved cobalt plumes. The other major subsurface source came from Upper
Labrador Seawater, which may carry higher cobalt concentrations due to the interaction of
this water mass with resuspended sediment at the western margin or from transport even
further upstream.  Minor sources of cobalt came from dust, coastal surface waters and
hydrothermal systems along the mid-Atlantic ridge. The full depth section of cobalt chemical
speciation revealed near complete complexation in surface waters, even with the regional high
dust deposition.  However, labile cobalt was found below the euphotic zone, demonstrating
that strong cobalt binding ligands were not present in excess of the total cobalt concentration
there and implying mesopelagic labile cobalt was sourced from the remineralization of
sinking organic matter.  Significant correlations were observed in the upper water column
between total cobalt and phosphate, and between labile cobalt and phosphate, demonstrating a
strong biological influence on cobalt cycling across much of the North Atlantic transect.
Along the western margin off the North American coast, this linear relationship with
phosphate was no longer observed and instead a relationship between cobalt and salinity was
observed, reflecting the importance of coastal input processes on cobalt distributions. In deep
waters, both total and labile cobalt were lower in concentration than at intermediate depths,
providing evidence that scavenging may remove labile cobalt from the water column. Total
and labile cobalt distributions were also compared to a previously published South Atlantic
GEOTRACES-compliant zonal transect (CoFeMUG, GAc01) to discern regional
biogeochemical differences. Together, these Atlantic sectional studies highlight the dynamic
ecological stoichiometry of total and labile cobalt.  As increasing anthropogenic use and





subsequent release of cobalt poses the potential to overpower natural cobalt signals in the
oceans, it is more important than ever to establish a baseline understanding of cobalt
distributions in the ocean.
**1 Introduction**
Cobalt is the scarcest of biologically utilized metals and has a complex marine
biogeochemical cycle. The small inventory of oceanic cobalt is maintained by a combination
of supply mechanisms, including sedimentary, aeolian, riverine/coastal, and hydrothermal
inputs. In particular, the high abundances of cobalt that have been observed as major plumes
within the low oxygen waters of major oxygen minimum zones of the South Atlantic and
South Pacific (Hawco et al., 2016; Noble et al., 2012), and from more limited datasets from
the North Pacific (Ahlgren et al., 2014; Saito et al., 2004; Saito et al., 2005), are likely due to
reductive dissolution and advection of sedimentary sources in regions with low-oxygen
bottom water sediment-water interfaces (Heggie and Lewis, 1984). Coastal and island sources
in oxygenated environments have also been observed, for example, off the North American
continental shelf (Saito and Moffett, 2002) and near the Kerguelen Islands (Bown et al.,
2012a). While there is limited information regarding the riverine and coastal fluxes of
particulate and dissolved cobalt to the oceans, earlier datasets show a significant "desorbable"
load of particulate cobalt as well as estuarine sources of organic cobalt complexes (Kharkar et
al., 1968; Zhang et al., 1990). The contribution of cobalt from dust has been more difficult to
directly observe because of the small amounts of cobalt present in dust relative to iron, with a
Co:Fe ratio in crustal material of 1:2600 (equivalent to 30ppm per 3-8% Fe by mass; Taylor
and McLennan, 1985), and its likely rapid utilization in the photic zone. Nevertheless,
laboratory and field studies have shown evidence for potentially significant dust contributions
to upper water column cobalt from anthropogenic and natural dust sources (Shelley et al.,
2012b; Thuróczy et al., 2010). While cobalt has been found to be enriched in end-member
hydrothermal fluids up to 2570 nM at TAG in the North Atlantic Mid-Atlantic Ridge (Metz
and Trefrey, 2000), input is thought to be relatively localized to near-vent environments due
to rapid removal by precipitating manganese and iron oxyhydroxides.
In addition to these natural sources, there has recently been tremendous growth in the
economic market for cobalt through the use of lithium batteries and their cobalt lithium oxide
cathode (Scrosati and Garche, 2010). This makes up a relatively new and large mobile



reservoir of cobalt throughout the world within electronics, homes, powerplants, cars, and
other devices. The environmental impacts of cobalt pollution via mining, smelting, and
inappropriate disposal of batteries will likely significantly increase in the future (Banza et al.,
2009). A baseline assessment of riverine and oceanic cobalt distributions is critical to inform
the development of sustainable economies with regard to trace metal biogeochemical cycles
and the future study of the industrial ecology of cobalt.
The chemical speciation of cobalt is dynamic in the oceans. Cobalt tends to be strongly bound
to organic complexes in the upper water column, but some fraction of dissolved cobalt
remains unbound or weakly bound as labile cobalt below the euphotic zone in intermediate
and deep waters. In the photic zone and upper water column, saturating concentrations of
cobalt binding ligands are often observed, particularly in oligotrophic regimes where
cyanobacteria are well represented (Saito and Moffett, 2001; Saito et al., 2005). These ligands
are extraordinarily strong, with conditional stability constants on the order of $>10^{16.8}$ (Bown et
al., 2012b; Saito et al., 2005), which is significantly higher than those for other transition
metals such as iron ($FeL_1$), for which measured stability constants are on the order of $10^{13.1}$
(Rue and Bruland, 1997, Buck et al., 2015). To achieve stability constants in this range, the
cobalt-ligand complexes almost certainly have a redox state of Co(III) (Baars and Croot,
2015; Saito and Moffett, 2001; Saito et al., 2005), which is consistent with Co(III) being a
textbook example of an inert metal redox state (Lippard and Berg, 1994). The structure of
cobalt ligands in seawater is currently unknown, but may be related to precursors or
degradation products of vitamin $B_{12}$, a cobalt-containing biomolecule. This complexation by
strong organic ligands likely protects and/or slows cobalt from scavenging (Saito and Moffett,
2001), as is similarly thought to occur for iron (Johnson et al., 1997). These ligands also have
a strong influence on the bioavailability of cobalt to microorganisms (Saito et al., 2002), and
the resultant microbial and phytoplankton ecology (Saito and Goepfert, 2008; Saito et al.,
2010; Sunda and Huntsman, 1995). While the strongly complexed cobalt is likely somewhat
protected from scavenging, the presence of labile cobalt in much of the oceanic water column,
should make that fraction particularly vulnerable to scavenging processes. Because
complexed and labile cobalt have different physicochemically driven cycles but are inherently
linked by biological transformations, cobalt speciation must be considered in efforts to fully
understand the biogeochemical cycling of cobalt in the oceans.





In recent years, there has been an emergence of ocean studies with high-throughput analyses
of dissolved cobalt (Bown et al., 2011; Bown et al., 2012b; Dulaquais et al., 2014a; Dulaquais
et al., 2014b; Noble et al., 2012) and labile cobalt (Noble et al., 2012), as a part of the prelude
to, or as part of, the international GEOTRACES program (Boyle et al., 2015). These studies
have considerably increased the available datasets on dissolved cobalt in the oceans and have
contributed to an understanding of cobalt cycling across several diverse biogeochemical
regimes. The Benguela Upwelling system appears to be a major source of dissolved Co to the
South Atlantic Ocean with a plume extending more than halfway across the basin. Cobalt was
also observed to be scavenged more slowly than other hybrid-type metals like Fe and Mn,
likely due to its slower oxidation kinetics and lower oxygen abundances in the oxygen
minimum zone (Noble et al., 2012). Additional datasets have explored the distribution and
speciation of cobalt in the Atlantic and Pacific sectors of the Southern Ocean (Bown et al.,
2011; Ellwood, 2008), the Ross Sea and McMurdo Sound of Antarctica (including seasonal
variability and under ice early spring conditions)(Noble et al., 2013; Saito et al., 2010), the
Eastern Tropical North Pacific and Costa Rica Dome (Ahlgren et al., 2014), the eastern
tropical North Atlantic (Baars and Croot, 2015), near the Bermuda, Hawaiian, and Kerguelen
Islands (Bown et al., 2012a; Noble et al., 2008; Shelley et al., 2012b), and throughout a
meridional transect of the western Atlantic Ocean (Dulaquais et al., 2014a; Dulaquais et al.,
2014b). The establishment of these high-throughput sampling and analytical methods for
cobalt, largely in response to the GEOTRACES program, has greatly improved our ability to
assess and monitor the biogeochemistry of this key micronutrient throughout the global
oceans. In fact, before 1990, there were fewer than 200 dissolved cobalt measurements
throughout the entirety of the oceans.
In this study, we examined the distributions of total dissolved cobalt and labile cobalt in the
North Atlantic during the U.S. GEOTRACES North Atlantic Transect (GA03/3_e). The
resulting ocean section from this study is compared to the GEOTRACES-compliant zonal
section in the South Atlantic Ocean (the CoFeMUG expedition, GAc01; (Noble et al., 2012)).
The North Atlantic is an ideal region for the study of biogeochemical processes given the
major contributions from aeolian dust deposition from the Sahara desert and Northern
hemisphere anthropogenic sources, proximity to coastal and continental sources, strong
hydrothermal sources associated with the mid-Atlantic ridge, and recently-formed North
Atlantic Deepwater. Moreover, based on previous studies in the South Atlantic, the
Mauritanian Upwelling region and the associated oxygen minimum zone were expected to



also exert strong influences on the distribution of cobalt in the ocean interior. Two companion
manuscripts describe this large dataset: the first describes the methodology, intercalibration
and preservation, oceanic distributions, chemical speciation, and major sources of cobalt to
the North Atlantic Ocean (Noble et al., this study). The second manuscript examines the
ecological stoichiometry of cobalt in zonal transects of the North and South Atlantic Ocean
(Saito et al., submitted).

## 2  Methods

Samples were collected along the U.S. GEOTRACES North Atlantic Transect, GA03/3_e
chief scientists: William Jenkins, Ed Boyle, and Greg Cutter). This transect was sampled in
two legs aboard the R/V Knorr: USGT10 (October 14, 2010 – November 3, 2010; GA03_e)
and USGT11 (November 4, 2011 – December 14, 2011; GA03). The first leg (USGT10)
departed from Lisbon, Portugal and followed a transect southward, sampling Mediterranean
Outflow Water (MOW) and concluding with a short westward transect along 17.4°N, crossing
the northern reach of the oxygen minimum zone associated with the Mauritanian Upwelling
system. This leg concluded at Station TENATSO at 24.5°W (USGT10-12). The second leg
(USGT11) departed from Woods Hole, MA and sampled stations along Line-W to the
Bermuda Atlantic Time Series (BATS) Station (USGT11-10, Fig. 1). At an approximately 3°
longitudinal spacing, the subsequent stations were sampled across the North Atlantic
Subtropical Gyre, including sampling the TAG hydrothermal plume and concluding with a
reoccupation of Station TENATSO (USGT11-24).

### 2.1  Sample Collection

Samples were collected using the Old Dominion University GEOTRACES Carousel on both
the 2010 expedition (USGT10) and the 2011 expedition (USGT11). Following the retrieval
of the carousel, the pre-conditioned, Teflon-coated Go-Flo bottles were moved to the
GEOTRACES Program class-100 trace metal clean van, pressurized with HEPA filtered air,
filtered through 0.2 μm Acropak filters in accordance with published methods (Cutter and
Bruland, 2012), and immediately refrigerated. Further information regarding deployment of
the GEOTRACES carousel can be found on the GEOTRACES website and in the
GEOTRACES cookbook (www.GEOTRACES.org). Samples were also collected using the
surface towed fish. These samples were collected by suspending the towed fish off the



starboard side with a boom, and sampled water at approximately 2m depth using a Teflon
diaphragm pump following the GEOTRACES Program Cookbook sampling
recommendations.
Sample storage bottles were prepared by soaking overnight in the acidic detergent, Citranox,
rinsed thoroughly with Milli-Q water (Millipore), filled with 10% HCl to soak for 10 days
(Baker Instra-analyzed HCl), rinsed thoroughly with Milli-Q water adjusted to pH 2, and
double-bagged, empty. Samples were kept in clean and rinsed 60mL LDPE bottles, and either
stored for a short time (<7 days) at 4˚C and double-bagged prior to analysis, or for a longer
time (6-40 days) at 4˚C, double-bagged with gas absorbing satchels and with the outer bag
heat-sealed to allow for longer-term sample preservation by removal of oxygen. The gas
absorbing satchels were iron-free, obtained from Mitsubishi Gas Chemical (model RP-3K),
and each satchel was rated to absorb 60mL of $O_2$ per 300mL of air. Each heat sealed bag
(Ampac™ Flexibles SealPAK Heavy-Duty Pouches, clear polyester 4.5mil) held 6-7 60mL
LDPE sample bottles and 3-4 gas absorbing satchels. The satchels come in impermeable,
vacuum sealed bags of 25. It would take a few days to use a full bag of 25 satchels, so the
bags were expended of air and re-heat-sealed after the required number of satchels were
removed in order to limit the exposure of the unused satchels to air. A heat sealer (Kapak by
Ampac) was used to seal each bag. After allowing the heat sealer to heat up for 3 minutes,
the bags were sealed by lining up the open ends of the bag in the heat sealer and sealing for 1-
2 seconds. When samples were ready to be analyzed, the bags were cut open and all samples
in the bag were analyzed within a week. Both labile and total dissolved cobalt were analyzed
from this sample bottle, and the sample identifier is the allocated GEOTRACES number.
### 2.2  Total dissolved and labile cobalt analyses
Concentrations of total dissolved and labile cobalt during USGT10 were determined
shipboard using a previously described cathodic stripping voltammetry (CSV) method (Saito
and Moffett, 2001; Saito et al., 2004). Measurements were made using the Eco-Chemie
µAutolabIII systems connected to Metrohm 663 VA Stands equipped with hanging mercury
drop electrodes and Teflon sampling cups within 7 days of sampling on double-bagged
samples that were kept in the dark at 4°C until analysis. Standard additions of cobalt were
carried out with Metrohm 765 Dosimats using a programmed dosing procedure (Noble et al.,
2008). Concentrations of total dissolved and labile cobalt from USGT11 were measured on
land between 1 and 6 weeks after the sampling date, using the same protocol as that employed



for USGT10. Analyses were performed on samples preserved in the dark and in gas
impermeable bags with gas absorbing satchels to ensure that no degradation of the sample
occurred during that time.
For total dissolved cobalt analyses, samples were UV-irradiated for 1 h prior to analysis using
a Metrohm 705 UV digester to degrade the organic ligands that bind cobalt and allow binding
by the added electroactive cobalt ligand, dimethylglyoxime. Samples were analyzed in 8.5
mL aliquots with the addition of 30 μL recrystalized dimethylglyoxime (DMG, Sigma-
Aldrich 0.1 mol L$^{-1}$ in methanol), 1.5 mL purified sodium nitrite (Fluka Analytical A.C.S.
reagent grade $\geq$ 99.0%, 1.5 mol L$^{-1}$ in Milli-Q water), and 50 μL purified N-(2-
hydroxyethyl)piperazine-N-(3-propanesulfonic acid) (EPPS) buffer (Sigma-Aldrich 0.5 mol
L-1 in Milli-Q water). Reagent purification protocols were modified from those previously
published (Saito and Moffett, 2001) in order to accommodate large batches. The DMG was
recrystallized after dissolving in an aqueous solution of EDTA to remove any traces of
metals. Nitrite solutions were prepared by equilibration overnight on a shaker with Chelex-
100 to remove any trace metals. The nitrite solution was then filtered and removed from the
chelex, the chelex was rinsed with copious amounts of milli-Q water, and added back to the
nitrite solution to equilibrate for a second night on the shaker, followed by filtration into acid-
washed HDPE or LDPE bottles. Nitrite was prepared in 500 mL batches and the batches were
blank-checked before shipping to sea. Cobalt concentrations were determined by the standard
additions technique, with initial concentrations measured in triplicate followed by four 25
pmol L$^{-1}$ cobalt additions. A 0.01 mM Co stock solution was prepared by the addition of 14.7
μL of a 1000 ppm cobalt standard to a 25 mL Teflon volumetric flask of Milli-Q water
adjusted to pH 2. 100 μL batches of 5 nM Co dosing solutions were prepared by the addition
of 50 μL of the stock solution to a 100mL HDPE trace metal cleaned volumetric flask of
Milli-Q water that was adjusted to pH 3 using Whatman pH indicator paper. This dosing
solution was added to the Dosimats and used for the standard additions. Final concentration
calculations were adjusted for dilution by the nitrite addition.
The analytical blank was determined by analyzing low-trace metal concentration seawater that
had been UV-irradiated for 1 h, equilibrated overnight with prepared Chelex 100 resin beads
(Bio-Rad), and UV-irradiated a second time to degrade any leached synthetic ligands. This
metal free seawater was kept at room temperature in trace metal cleaned Teflon bottles of
250ml and 500ml capacity. Blanks for each reagent batch (nitrite, DMG, EPPS) were





subtracted from the initial sample concentration. Blank analyses for each reagent batch were
made at the beginning and end of use to confirm that the blank remained constant during
analyses. The averaged blank for all reagent batches for the entire dataset was 4pM ± 1.2 with
a range of 1.7 - 6.3pM (n = 38 for individual blank analyses). For a given reagent batch, the
standard deviation was smaller, and we report a detection limit (3 times the standard deviation
of the blank) of 1.8pM, representing the average of the detection limits estimates for reagent
batches with at least 3 blank analyses (n = 6).
For labile cobalt analyses, 8.5 mL of sample was pipetted into acid washed Teflon vials that
were preconditioned with a small aliquot of sample water. 30 μL aliquots of DMG were
added to each vial and allowed to equilibrate overnight in the dark prior to analysis (Saito et
al., 2004). Analyses were then performed as described for total concentrations using the
standard addition technique with the addition of the remaining two reagents immediately
before analysis. Previously, we determined that natural cobalt is strongly bound to ligands in
seawater with a conditional stability constant of $>10^{16.8}$ (Saito et al., 2005). We define labile
cobalt (LCo) as the fraction of total dissolved cobalt (dCo) that is exchangeable with the
DMG complexing agent, indicating it is either bound to weak organic/inorganic ligands in
seawater or present as free Co(II) (Saito et al., 2004; Saito et al., 2005).  Where labile cobalt
is detectable, the strong cobalt ligand concentration is defined as the difference between the
total dissolved cobalt and the labile cobalt.
Two full electrochemical systems were utilized for analyses, with one dedicated to total cobalt
analyses and the other to labile cobalt analyses. GEOTRACES standard seawater and internal
standard lab seawater were analyzed periodically to ensure that the two electrodes were
intercalibrated and functioning properly (Table 1). GEOTRACES standard seawater was UV
irradiated and neutralized using 1N Optima ammonium hydroxide to increase the pH to 7.5.
An oligotrophic seawater standard internal to our lab (described as CSW for consolidated
seawater standard in Table 1), was prepared by UV irradiation in 500mL batches and stored in
trace metal clean Teflon bottles at room temperature. The standard was not acidified at any
point, thus avoiding the introduction of error and reagent blank associated with adding acid
and base (Saito and Moffett, 2002), and allowing regular re-analysis without any further
treatment. These standards were used to troubleshoot when the electrodes malfunctioned and
to ensure consistency when operational. These batches were measured to be 53 ± 3 pM (n =
4), 74 ± 3 (n = 9), 71 ± 3 pM (n = 16) and 54 ± 6 pM (n = 35) over the course of the USGT-11



cruise analyses, and 53 ± 5pM (n = 24) for the USGT-10 cruise analyses, measured across all
reagent batches and both electrode systems. These results demonstrate that the methodologies
employed to produce this dataset detect concentrations within the standard deviation of
current consensus values for UV-irradiated samples, which can be found on the International
GEOTRACES Program website (www.geotraces.org, see below). On occasion analyses were
repeated due to obvious electrode malfunction or to confirm oceanographic consistency of
measured values. If the repeated measurement was similar to the initial measured value, the
initial value is reported. If the repeated analysis was more oceanographically consistent with
adjacent values in the water column, that analysis was used instead.

### 2.3 Intercalibration efforts and data repository

Our laboratory has participated in the GEOTRACES intercalibration effort using this
electrochemical analytical technique. We report our laboratory values for the GEOTRACES
and SAFe standard analyses using the above-described electrochemical technique, including
those conducted during analysis of the US North Atlantic GEOTRACES Section samples to
be: SAFe S1 = 5.4 ± 2.6 (n = 9), SAFe D2 = 48.3 ± 5.5 (n=7), GEOTRACES GS = 31.4 ± 4.1
(n = 24), GEOTRACES GD = 66.9 ± 6.2 (n = 30). These results are in good agreement with
those from the GEOTRACES intercalibration effort for Co using different methods all using
UV-oxidation to degrade strong cobalt ligands.
Comparisons of our CSV data with ICP-MS and flow injection methods at the Bermuda
Atlantic Time Series station, a crossover GEOTRACES station, from this expedition and the
Dutch GEOTRACES section GA02, generated values that were similar in the photic zone but
higher than others' studies in the mesopelagic (Dulaquais Refs). These observations were
reported to the GEOTRACES Intercalibration committee and have delayed incorporation of
Atlantic dissolved cobalt data into the Intermediate Data Products. The higher mesopelagic
values we observed on fresh and "gas-satchel" preserved samples appear to be real based on
comparisons with our own unpreserved samples (see Section 3.3 below)
All data generated by this lab and discussed in this paper have been submitted to BCO-DMO
and are available at <http://www.bco-dmo.org/dataset/3868>. If using this data for future
publication or analyses, please cite: Saito, M. (2013) Total dissolved Cobalt and labile Cobalt
concentrations from R/V Knorr cruises KN199-04 and KN204-01 in the Subtropical northern
Atlantic Ocean from 2010-2011 (U.S. GEOTRACES NAT project). Biological and Chemical





Oceanography Data Management Office (BCO-DMO). Dataset version: 2013-04-26. URL:
http://www.bco-dmo.org/dataset/3868.
**3  Results and Discussion**
**3.1  Oceanographic Setting**
The US GEOTRACES North Atlantic expedition track (Fig. 1) was chosen to investigate
multiple processes and provinces within the constraints of an approximately zonal section and
was completed in two legs. USGT10 sampled from Portugal to the Cape Verde Islands and
consisted of stations USGT10-01 to USGT10-12 (October 2010), and USGT11 sampled from
Woods Hole, MA, USA to the Cape Verde Islands, and consisted of stations USGT11-01 to
USGT11-24 (November-December 2011).  From west to east, the track transited from
seasonally productive New England coastal waters, to shelf and slope waters, crossing the
deep western boundary current (DWBC) and the Gulf Stream along the repeat hydrography
section, Line W. After occupying the Bermuda Atlantic Time Series station (BATS), the track
crossed through the oligotrophic Sargasso Sea, the North Atlantic Subtropical Gyre, and the
Mid-Atlantic Ridge hydrothermal Trans-Atlantic Geotraverse (TAG) site. From there, the
transect continued east, traversing over the northern reach of the tropical North Atlantic
Oxygen Minimum Zone near the Cape Verde Islands and Mauritanian Upwelling region off
the coast of Northwest Africa. The most eastward samples collected were along a meridional
section from Portugal to the Mauritanian Upwelling, and sampled Mediterranean Outflow
Water (MOW).
**3.2  Vertical Profiles and Sections of Total Dissolved Cobalt and Labile**
**Cobalt in the North Atlantic Ocean**
The dissolved cobalt data product from USGT10 and USGT11 consisted of 11 and 21
profiles, respectively, totalling 717 total dissolved cobalt and 717 labile cobalt data points that
were compiled into ocean sections that were rendered with Ocean Data View (Figs 2, 3A-B,
Schlitzer, 2011).  Visual examination of the vertical profiles (Fig 2) and sections (Fig 3)
showed oceanographically consistent results. The two expeditions included a repeat
occupation of a station at TENATSO (Tropical Eastern North Atlantic Time-Series
Observatory, USGT10-12 and USGT11-24) where the mean difference between the two
profiles was 8 pM overall, and  2.2 pM below 2000m.  These profiles are expected to be



similar given the slow timescale of deeper watermass movement relative to the 1 year
sampling interval, and this resampling provides a unique opportunity to examine temporal
variability on this time scale throughout the water column.  Larger differences were observed
in the surface and mesopelagic waters, with little to no difference below 2000m depth.  In
surface waters, this difference is explained by the fast movement of surface waters and
timescales of the processes affecting cobalt cycling in this highly biologically active part of
the water column.  In mesopelagic waters (particularly between 1500 – 2000m), differences
were observed within a water mass characterized as >55% Upper Circumpolar Deepwater
(UCPDW) *via* OMPA analysis (Jenkins et al., 2015), indicating some temporal variability at
these depths even within the relatively short time scale between samplings.
### 3.3 Preservation and accuracy of total dissolved cobalt using gas
### absorption satchels
The measurement of total dissolved cobalt has always been a challenge due it having the
lowest concentrations of any biologically used metal. During the CoFeMUG expedition slight
differences between at-sea analyses and samples returned to the laboratory were observed
within the cobalt maximum inside the oxygen minimum zone that raised suspicions of a
preservation issue for dissolved cobalt (Noble and Saito, unpubl. data). Moreover, during
GEOTRACES intercalibration efforts, two issues have arisen that also contribute to this
difficulty in accurate and reproducible total dissolved cobalt measurements. First, during the
initial GEOTRACES intercalibration effort, it was confirmed that UV-irradiation was
required in all methods to release cobalt from organic ligands that do not degrade or
dissociate bound cobalt at low pH. This lack of sensitivity of cobalt ligands to dissociation at
low pH is not surprising: it is well known that the cobalt-containing biomolecule vitamin $B_{12}$
survives similarly low pH in the human stomach without dissociation, and that Co(III)
complexes are classic examples of kinetic inertness and stability (Lippard and Berg, 1994).
As a result, all samples reported as total dissolved cobalt here and in all of our previous
studies have been UV-irradiated. More recently, we have become concerned that some
intermediate depth samples are prone to loss of dissolved cobalt during storage via redox or
other unknown reactions. To document this phenomenon, we present three full profile repeat
analyses of  USGT10-9 off the coast of Mauritania, analyzed by three preservation protocols:
A) at-sea analyses performed within 2 days of sample collection, B) in-lab analyses
performed after four months of storage at 4° C in the dark, and C) in-lab analyses of sample





duplicates after four months of storage at 4° C in the dark, where the bottles were additionally
preserved in air-tight, heat-sealed bags with gas absorbing satchels (Fig. 4). Seawater for the
unpreserved, stored analyses (B above) was taken from the same bottles as the at-sea
analyses, thus the sample bottles had a large headspace (60 mL bottles with ~50% headspace).
These analyses and the at-sea analyses showed similar dissolved cobalt concentrations at the
top and base of the water column but showed a large deviation at all other depths (Fig. 4). Use
of the gas absorbing satchels to store samples for the same length of time (C above) allowed
for excellent recovery of dissolved cobalt with a slope close to the 1:1 coherence of 0.96 and
an $r^2$ of 0.99. Interestingly, almost all of the labile cobalt measured at sea had disappeared in
unpreserved samples, indicating the movement of cobalt between chemical forms on the
timescale of these experiments (data not shown; Noble and Saito in prep).  This has major
implications for cobalt speciation on preserved samples in certain biogeochemical regimes,
especially the North Atlantic. Interestingly, samples from the Ross Sea did not experience this
loss, showing excellent reproducibility on stored, unpreserved samples for both total
dissolved and labile cobalt after 17 months (Noble and Saito in prep).  However, given the
successful recovery of total cobalt demonstrated by this new technique in a region prone to
low oxygen and heavy dust inputs, we encourage research groups measuring dissolved cobalt
to adopt the preservation method used in this study.
A GEOTRACES crossover station was also included at the Bermuda Atlantic Time-Series
Station (BATS, USGT11-10). Data were compared between our lab and two labs that relied
on an ICP-MS method (Middag *et al*. 2015) at this station as well as at a second station in the
North Atlantic Subtropical Gyre (USGT11-20).  Our laboratory results were found to be
consistently higher than those of the other groups (intercomparison data not shown).  The
largest discrepancies (~20pM) were observed at intermediate depths associated with the
highest labile cobalt concentrations (up to 36pM). Discrepancies were generally smaller in
deeper waters (where labile concentrations were often ≤10pM), and concentrations were often
within a few pM of each other in the upper few hundred meters where labile concentrations
were below 10pM and often below our detection limit. Based on our comparison and
preservation experiments in this and other locations, the preservation and storage issue
appears to be exacerbated in the North Atlantic, and has only a minor influence at some
depths in the South Atlantic, Ross Sea, and South Pacific.  Hence, we hypothesize that the
preservation effects may be related to the extensive dust- and subsequent colloidal-loading of
the North Atlantic region.  Ultimately, because comparison of our method with GEOTRACES



standards and our internal laboratory standard showed excellent accuracy and reproducibility
(see Table 1), we interpret our higher concentrations at intermediate depths to be due to loss
of cobalt associated with different preservation techniques used in other methods.. Again, this
preservation effect appears to be strongest in the North Atlantic, demonstrating only a minor
influence in other regions.

### 3.4  Major sources of cobalt to the North Atlantic Ocean

The dissolved cobalt data highlight continental margin sources of cobalt to the intermediate
waters of the North Atlantic from both eastern and western margins: a large plume emanated
from the African coast along the eastern margin (Section 3.4.1), and another large plume was
observed along the western margin within Upper Labrador Seawater (ULSW) (Section 3.4.2).
In addition, regional contributions from coastal inputs (Section 3.4.3) were observed and a
small, localized plume of cobalt was detected above the Mid-Atlantic Ridge hydrothermal
vent site at TAG (Section 3.4.5). Atmospheric deposition over the tropical and subtropical
North Atlantic is a significant source of a number of metals (e.g. Fe and Mn), but trace cobalt
appears to be only a small contribution to the water column inventory. Notably, all elevated
source signals of total dissolved cobalt were coincident with elevated labile cobalt as well.
While the magnitude of the signals differed between the two species, this elevated signal
coincidence may indicate sources carried within a water parcel that experiences slower
scavenging relative to surrounding waters (*e.g.* due to low oxygen concentrations) or that the
inputs were relatively recent and the maxima were captured in this sampling effort before they
were fully scavenged (*e.g.* close to hydrothermal inputs). The following sections discuss these
cobalt sources to the North Atlantic Ocean and compare the relative magnitudes of those
sources to those observed in a prior study of the South Atlantic Ocean (Noble et al., 2012).

### 3.4.1 A large plume of cobalt off the Mauritanian Coast

The largest feature of this dataset was the dissolved cobalt plume observed along the eastern
margin off of North Africa (Fig. 3). This subsurface plume of dissolved and labile cobalt
extended from the Mauritanian coast more than 2000 km into the basin, based on a
conservative definition of the plume of exceeding 100 pM total dissolved cobalt (Fig. 3).
Centered around the oxygen minimum zone, the highest concentrations of dissolved cobalt
(160 pM) were detected at ~400 m depth and were primarily associated with Atlantic
Equatorial Waters (AEW). Wind- and circulation-driven upwelling occurs along the





Mauritanian coast, leading to higher overall productivity that supports important local
fisheries. The subsequent substantial remineralization of organic matter contributes to low
oxygen waters at intermediate depths. This Northwest African/Mauritanian Upwelling region
contains the smallest of five major marine oxygen minimum zones in the oceans, with the
others located in the Eastern Tropical North and South Pacific, the Eastern South Atlantic,
and the Arabian Sea (Keeling et al., 2010). Previous cobalt studies have shown that the South
Atlantic OMZ and the two Pacific OMZs all harbor high concentrations of cobalt (Ahlgren et
al., 2014; Hawco et al., submitted; Noble et al., 2012; Saito et al., 2004; Saito et al., 2005).
The current study confirms high cobalt concentrations in the North Atlantic oxygen minimum
zone as well, despite this OMZ having higher $O_2$ concentrations and lacking the substantial
suboxic and anoxic waters found in other OMZs.
The elevated cobalt observed in the Mauritanian Upwelling is due to a combination of
processes: 1) low bottom water oxygen allowing reductively dissolved cobalt to escape from
sediments and be transported long distances with minimal removal,, and 2) the poorly
ventilated shadow zone waters of the OMZ allowing accumulation of cobalt from vertical
export of remineralized biogenic and aeolian cobalt.  Similar coupling of processes and
elevated cobalt were observed in the South Atlantic OMZ on a GEOTRACES-complaint
zonal section (GAc01 also known as CoFeMUG, (Noble et al., 2012). These two parallel
transects afford a unique opportunity to compare contributions from multiple sources that
result in similar large-scale dissolved cobalt features. The biogeochemistries of the two
regions are somewhat distinct: the North Atlantic is heavily influenced by aeolian input from
the Sahara Desert and North America, and upwelling off the coast of Mauritania is ~1.8 Sv
according to [3]He measurements (Jenkins et al., 2015). In contrast, the South Atlantic
experiences very low overall dust deposition, and upwelling in the Angola dome and
Benguela Upwelling has been estimated to be 2.2 Sv (Frame et al., 2014; Skogen, 1999). In
the South Atlantic, the cobalt plume was also centered around the oxygen minimum and was
coincident with elevated dissolved manganese and iron (Noble *et al*. 2012). Similarly,
elevated manganese and iron were observed coincident with the North Atlantic OMZ,
suggestive of similar processes influencing these trends (Hatta et al., 2015).  In the South
Atlantic, we suggested that the high OMZ concentrations of these hybrid metals, and cobalt in
particular, were due to a combination of reductive dissolution, upwelling, advection, and
remineralization (Noble et al., 2012). Reductive dissolution can be a source of cobalt *via*
release of cobalt associated with manganese oxides in sediments along the coast, as we



previously suggested for the South Atlantic OMZ system (process #1 above). This process
likely contributes to the plume in the North Atlantic; however, the fraction of the cobalt
plume supported by aeolian contributions to the vertical export (process #2 above) would be
expected to be higher. Moreover, oxygen concentrations in the North Atlantic are not as low
as those observed in the South Atlantic, but particulate $FeS_2$ has been observed in both the
sediments and suspended particulate matter near the Mauritanian Upwelling sampling sites
(Lam et al., 2012), suggesting that there may be sufficiently low oxygen concentrations along
the shelf to allow the escape of reduced cobalt from the sediments without reprecipitation as
oxides. Despite higher mesopelagic oxygen concentrations in the North Atlantic, the
dissolved cobalt concentrations were also higher here, likely due to a larger contribution from
dust sources in the North Atlantic study area (see Section 3.4.4) and/or through less time
exposed to scavenging processes within the ocean interior.   With the addition of the
GA03/3_e section in the North Atlantic, four of the five world's major coastal OMZ regions
have now been found to harbour high concentrations of cobalt (Hawco et al., submitted;
Noble et al., 2012; Saito et al., 2005).   This adds to the growing evidence that oxygen
minimum zones and their accompanying coastal regions are important sources of dissolved
cobalt to the oceans. Importantly however, for a basin-scale plume to be observed in the
North African OMZ region implies that cobalt plume formation and persistence by slowed
scavenging has a higher (low) oxygen threshold than other OMZ processes (e.g.
denitrification) that require suboxic or anoxic conditions.

21       ### 3.4.2 Advected sedimentary source from Upper Labrador Seawater

Strong total dissolved cobalt and labile cobalt plumes were also observed in the western
Atlantic along Line W (USGT11-01 to the Bermuda Atlantic Time Series station (BATS,
USGT11-10) between 1000-1500m depth, with no accompanying low oxygen signal.  Water
mass analyses using Optimum Multi-Parameter Analysis (OMPA, Jenkins et al., 2015)
constrains the dissolved cobalt feature to be contained within Upper Labrador Sea Water
(ULSW). Low silicate concentrations are a differentiating feature of ULSW and can be used
to illustrate this by overlaying silicate contours on a Western Margin section of dissolved
cobalt (Fig. 5). Two processes, which are not mutually exclusive, may explain the observed
feature: 1) advection of a water mass that contains a higher inventory of cobalt than the
surrounding waters and/or 2) coastal input of cobalt released from shelf sediments as ULSW
comes in contact with the coastal shelf and slope during the transit south.  Hatta *et al*. (2015)



observed high Fe and Mn on this same GEOTRACES transect within this water mass and
invoked release of these metals from sediments into the water column. Previous studies have
also invoked continental margin interaction to explain Fe enrichment in Labrador Sea Water
at a station further northeast into the Atlantic basin (Laes et al. 2003), and recent data suggests
that Arctic waters may contain very elevated concentrations of cobalt (Saito and Noble
unpublished data, Bundie and Saito, unpublished data) which could provide a source of high
cobalt to the locations of ULSW ventilation.
The ULSW cobalt plume appears to be different in composition from that of the eastern
margin Mauritanian Upwelling feature.  First, the percentage of labile cobalt is higher (35-
40%) in the western margin ULSW feature than within the eastern margin Mauritanian
Upwelling (20-25%).  Higher particulate cobalt is also observed along the western margin
(Fig. 3), and could be related to the higher abundance of both dissolved phases, reflecting
increased interaction with this phase via shelf inputs and/or scavenging. The transport of
labile cobalt to depths below the photic zone may prevent entrainment of labile cobalt into
microbial cycling and its transformation to complexed cobalt. This could explain the
speciation differences relative to the eastern basin where the plume is shallower and labile
cobalt is a smaller fraction of total cobalt.  Oxygen concentrations are also much higher in
ULSW than within the Mauritanian Upwelling plume, and further demonstrating that low
oxygen is not necessarily critical to sustaining subsurface cobalt plumes.  Another possible
contribution to the western margin plume could come from remineralized cobalt transported
within ULSW from its origin to the north. Cyanobacteria are thought to be major contributors
to the oceanic cobalt ligand inventory and their virtual absence in polar regions has been
invoked to explain the often higher fraction of labile cobalt found in the euphotic zone of
those regions (Noble et al., 2013; Saito et al., 2010). It appears that the cold polar-sourced
Labrador Sea waters may also carry that imprint of higher labile cobalt. These potential
contributions are not mutually exclusive: it is likely that both continental shelf inputs and
advected remineralization signals from cooler regions contribute to this high cobalt feature of
the North American continental shelf and slope environment.

### 3.4.3 Coastal sources along the North American Margin

In the upper 40-60m along the western margin, surface coastal sources dominate cobalt
distributions, and an inverse linear relationship with salinity is observed, indicative of a
freshwater endmember source (Fig. 6).  While biological processes often drive relationships



of cobalt with phosphate instead of salinity, including at most of the stations sampled during
GA03/3_e (see section 3.7), these Co:PO$_4^{3-}$ correlations (Co:P hereon) were absent in the
Line-W region (Fig. 6). Previous work characterized a similar relationship between salinity
and cobalt in the North American margin region (Saito and Moffett, 2002), as well as between
salinity and other elements such as copper and nickel (Bruland, 1980).  In the current study,
the relationship with salinity was similar for labile cobalt, supporting a labile source from the
coast.  This input of cobalt in conjunction with lower salinities implies potential sources from
freshwater input such as rivers or groundwater from the coastal Atlantic region.

### 3.4.4 Evaluating Aeolian sources to the North Atlantic

The North Atlantic Ocean is strongly influenced by atmospheric dust deposition,
which provides an important source of iron and other metals and has been shown to have
impacts on regional nitrogen fixation (Moore et al., 2009). The influence of aeolian input
from the Sahara Desert increases moving eastward toward the North African margin
(Mahowald et al., 2005; Shelley et al., 2012a, Ohnemus and Lam, 2015). The Sahara is an
important source of iron and other metals to the North East Atlantic (Measures 1995;
Measures et al. 2008, Shelley et al. 2015). The two legs of the GA03/3_e section both
occurred during autumn/winter months in successive years (October-December 2010, 2011),
coinciding with the period of lower atmospheric deposition to the western Atlantic as
measured at Bermuda, where spring is the major period of deposition (Engelstaedter et al.,
2006; Jickells et al., 1990). Dust samples collected on-board this North Atlantic section
(GA03/3_e) showed aerosol cobalt loadings associated with lithogenic elements such as Ti,
Al, and Fe, with minor contributions from other aerosol types (Shelley et al., 2015),
suggesting that desert dust sources were more significant than anthropogenic sources during
the GA03/3_e expeditions. Lithogenic dust sources are likely a less significant source of
cobalt to the North Atlantic Ocean than they are for other metals such as iron and aluminum
because cobalt is much less abundant in crustal material (average Co:Fe ratio of ~1:2600,
Taylor and McLennan, 1985), resulting in aeolian influences competing with the large coastal
cobalt sources described above. In this section, we examine the contribution of dust to
dissolved cobalt inventories using cobalt distributions across the basin, correlations with
dissolved aluminum in the eastern basin, and estimates of dust flux contributions and relative
to upwelling fluxes.



Unlike iron or aluminum profiles, which show persistent surface maxima on GA03/3_e,
dissolved cobalt profiles within the upper water column of the oligotrophic gyre were
consistently nutrient-like (Fig. 2), with surface concentrations of total cobalt as low as 9 pM
due to biological uptake (Fig. 7A-B). This was consistent with the low-dust sampling timing
and prior observations. Intriguingly, previously published profiles of total dissolved cobalt at
BATS station have displayed either nutrient-like or atmospheric deposition surface
maximum-type depending on the time of year sampled, the seasonality of dust deposition
(atmospheric deposition is highest during late spring/ early summer), and mixed layer depth
(Saito and Moffett, 2002; Shelley et al., 2012b). The absence of a surface maxima at BATS
during GA03 was consistent with low dust fluxes during fall and winter months, and with the
deepening of the mixed layer, which acts to dilute dust-borne cobalt dissolved into shallow
mixed layers during summer.
Dust deposition to the oligotrophic gyre appeared to have a small impact on the surface cobalt
inventory during the low fall/winter dust flux and deep seasonal mixed layer sampled by the
GA03 expedition. Aerosol cobalt deposition near Bermuda can be estimated as the product of
aerosol cobalt concentrations determined from shipboard bulk aerosol sampling (0.15 pmol m$^{-}$
$^{3}$ for BATS and 0.44 pmol +/- 0.28 m$^{-3}$ including 12 surrounding deployments to BATS,"
BATS region" hereon (Shelley et al., 2015)) and a typical dry deposition rate of those aerosols
(1000 m / d for the Bermuda Atlantic Time-Series crossover station, Duce et al. 1991). This
results in a cobalt deposition flux of 0.15 nmol / m$^2$ / d (BATS) and 0.44 nmol / m$^2$ / d (BATS
region). The solubility of cobalt in Saharan-derived aerosols collected in the Sargasso Sea has
been estimated to be 10% during periods of high dust deposition (Shelley et al., 2012a),
similar to longer term dissolution experiments on lithogenic-rich aerosols collected in the Red
Sea (17 +/- 7%, Mackey et al., 2014). Combining these three facets (flux, solubility and
mixed layer depth) with an observed mixed layer depth of 80m observed during GA03, dust
dissolution is estimated to add ~0.06 (BATS) to 0.17 (BATS region) pM Co per month. This
was a relatively low flux compared to the measured mixed layer inventory of 30 pM at BATS
during GA03 for this period of low dust deposition and deep mixed layers.
We can also consider the conditions of high dust deposition coincident with the seasonally
stratified mixed layer to capture the maximum potential dust contributions to the shallow
cobalt inventory. Dust deposition and mixed layer depth at BATS tend to experience strong
seasonality with similar phasing (low dust coincident with deep mixed layers and high dust



deposition with shallow mixed layers). During the summer when dust fluxes are highest,
mixed layers can be < 10 m deep (Steinberg *et al*. 2001). Shallow mixed layers intensify the
assimilation of metals from atmospheric deposition because the fluxes are diluted over a
smaller volume (Jickells 1999). Annual aerosol cobalt fluxes at BATS were calculated to be
944 nmol Co m$^{-2}$ year$^{-1}$ using $^{7}$Be isotopes and data from July 2011 to June 2012 (Kadko et
al., 2015). Considering the extreme case where 100% of this annual dust deposition is
deposited under highly stratified summer conditions (10m mixed layer depth), with an
assumed 10% Co solubility results in an estimated 9.4 pM increase per year to the mixed
layer cobalt inventory. This is a potentially significant contribution compared to the dissolved
cobalt observed during GA03 (9-36 pM). Moreover, solubility increases with seawater
exposure time (Mackey et al., 2014), episodic dust loadings of high intensity, and/or an
increased anthropogenic component with higher solubility (Thuróczy et al., 2010) could also
enhance the fractional magnitude of aeolian sourced dust to the mixed layer. As a result,
higher dust deposition and shallower mixed layer depths that occur in the spring and summer
at BATS could explain the non-nutrient-like profiles previously observed (Hansell and
Carlson, 2001; Saito and Moffett, 2002; Shelley et al., 2012b).
Taken together, these results imply that the strong seasonal cycle at the BATS station creates
a dichotomy of aeolian influences on the cobalt inventory in mixed layer. As winter time
convective overturning homogenizes the upper water column, the spring and summer dust
deposition becomes diluted. When applied to a 100m mixed layer, the same annual dust
deposition flux decreases to 0.9 pM per year, increasing the dissolved inventory by a few
percent overall (2.5-10% of the range described above). As dissolved cobalt concentrations
increase with depth, winter mixing also provides a considerable flux of cobalt to the surface
from deeper waters (Saito and Moffett, 2002). As a result, atmospheric cobalt deposition is
most apparent in the mixed layer cobalt inventories on seasonal timescales in the North
Atlantic.
Despite the expectation that dust sources of cobalt from the Sahara Desert may have an even
stronger influence on eastern margin cobalt inventories, sedimentary sources appear to be
dominant. Closer to the Saharan dust sources to the east, increasing surface cobalt
concentrations were observed on GA03/3_e (Fig. 7A-B). Previously, surface cobalt
concentrations of up to 110 pM (higher than those observed on GA03/3_e) were measured in
this region and attributed to dissolution of Saharan aerosols (Bowie *et al*. 2002). Yet, because



both dust deposition and coastal upwelling occur in this region, elevated surface
concentrations near the eastern margin cannot be solely attributed to dust deposition.
Similarly elevated dissolved cobalt was also observed near the coastal margin across the
South Atlantic zonal transect, which experiences much lower dust inputs than the North
Atlantic (Fig. 7C). Moreover, eastern margin profiles of dissolved cobalt from both
expeditions were similar in structure and concentration despite major differences in dust
supply and a closer proximity of North Atlantic margin profiles to the coast than those in the
South Atlantic, implying that sedimentary sources were dominant in both OMZ regions (Figs.
7E-F, see caption).
Upwelling can be demonstrated to be the major source of cobalt to the euphotic zone in the
North East Atlantic. Similar to the estimates at BATS, the contribution of aerosol dust to
dissolved cobalt can be estimated from the aerosol cobalt concentrations measured on
GA03/3_e (Shelley et al., 2015), standard deposition velocities (Duce et al., 1991), and
relative solubility (Shelley et al., 2012). Dust inventories in North African dominated aerosols
along USGT10 averaged 17 pmol Co $m^{-3}$ (Shelley et al., 2015), over 100-fold higher than that
measured at BATS (0.15 pmol Co $m^{3-}$, discussed above). These measurements imply a
soluble cobalt flux on the order of 1.7 nmol $m^{-2}$ $d^{-1}$ to the mixed layer between USGT10-08
and USGT10-12. Due to the elevated cobalt in the OMZ plume described above, upwelling
fluxes of dissolved cobalt to the mixed layer are significant here. Jenkins et al. 2015 estimated
upwelling rates during GA03/3_e to be 5 m $d^{-1}$ (Jenkins et al., 2015). A dissolved cobalt
concentration at the base of the mixed layer of ~50 pM implies an upward flux 250 nmol $m^{-2}$
$d^{-1}$. As a result, the soluble cobalt flux from dust during GA03/3_e (1.7 nmol $m^{-2}$ $d^{-1}$) is only
~1% of the upwelling flux (250 nmol $m^{-2}$ $d^{-1}$). However, Ohnemus and Lam also observed
that the particulate material found in the mesopelagic of this region had a strong lithogenic
signal that they attributed to dust fluxes, presumably through sinking dust material (Ohnemus
and Lam, 2015), raising the possibility for a gradual dissolution of cobalt from sinking
lithogenic particles, as observed experimentally by Mackey et al. (2014). Overall, these flux
calculations show that the sub-surface cobalt inventory observed on the eastern portion of this
transect buffered the cobalt inventory from dynamic dust deposition, although there may also
be a component of dust supply within this subsurface inventory with particle sinking and
dissolution.



Tracers of dust input can help distinguish external sources of cobalt from dust, and
comparisons between dCo:dAl in the North and South Atlantic surface waters show
significant differences.  Despite the dominance of upwelling fluxes of cobalt in the Eastern
North Atlantic, dissolved cobalt was observed to correlate with dissolved aluminum, a tracer
of lithogenic dust deposition in surface waters between USGT10-08 and USGT10-12 ($r^2 =$
0.96, Fig. 7, dissolved aluminum data from Measures et al. (Measures et al., 2015)). The slope
of this relationship (1-2 mmol dCo: mol dAl) was much steeper than that expected from their
relative abundance in aerosols on GA03/3_e (0.16 mmol Co : mol dAl), despite their similar
solubilities in North African aerosols (5-15% Buck *et al*. 2010; Mackey *et al*. 2015; Shelley *et*
*al*. 2012b). Perhaps this deviation was related to artifacts in solubility measurements or
differential biological processing: productivity is quite high in the Mauritanian Upwelling
region and this dCo:dAl relationship in surface waters may reflect rapid uptake of both
elements  (biological uptake for cobalt and scavenging for aluminum) and subsequent release
by remineralization. This influence was evident in depletion of both elements in the upper
water column (Figs. 2, 3), and in dAl's lower abundance in the eastern (near Africa) portion
of the GA03/3_e transect relative to the west (near BATS, (Measures et al., 2015)).
Intriguingly, this dCo:dAl relationship was not observed in surface waters of the Benguela
Upwelling where dust input was much lower, but upwelling was also strong (Noble et al.,
2012). This coupling of dCo and dAl in the North Atlantic implies both an influence of dust
and a complex interaction with the high productivity of the upwelling region.
As demonstrated above, high cobalt concentrations in the underlying OMZ cause upwelling
fluxes of cobalt to be much larger than dust dissolution. Since cobalt in the surface ocean is
acquired by phytoplankton, exported to depth and then remineralized, it is possible that
atmospheric deposition of cobalt contributes to the OMZ cobalt plume indirectly, thereby
returning to the surface ocean when these waters are upwelled. Tritium/helium ages of these
water masses have ventilation ages on the order of several decades (Jenkins et al. 2015),
allowing cobalt originally delivered to the surface ocean to accumulate in the OMZ after it is
remineralized. Continued dissolution of cobalt from dusts that have already sunk below the
euphotic zone may provide an additional cobalt source to these depths (Mackey et al. 2014).
Therefore, despite instantaneous dust fluxes that are dwarfed by ocean mixing, storage of
dust-derived cobalt in the mesopelagic ocean may cause dust-borne cobalt to be significant in
sustaining the cobalt inventory in the North Atlantic Ocean on longer timescales.





### 3.4.5 **Hydrothermal source of cobalt to the deep North Atlantic**

The influence of hydrothermalism on dissolved and particulate cobalt was clearly detectable

in near-field mid-Atlantic ridge samples, but unlike iron and manganese, these effects did not

persist appreciably beyond the ridge station. In the North Atlantic deep water concentrations

of cobalt were low (39-55 pM), likely due to scavenging and the intrusion of deep water

masses with a smaller cobalt inventory. Samples taken at the TAG hydrothermal field

(USGT11-16), however, showed a subtle increase in cobalt concentration relative to the

surrounding waters (Fig. 2, 3, 8). Five samples were taken within the plume between 3200

and 3500m depth above the well-studied TAG hydrothermal vent site (Fig. 8). A maximum in

both dissolved and labile cobalt was observed, constituting a ~37% increase over adjacent

depths within the vertical profile for a ~25pM hydrothermal signal over background of which

~16pM is labile cobalt. The presence of labile cobalt in the hydrothermal plume implies that

cobalt was released primarily in a labile form and that the vent may act as a local but small

source of cobalt to surrounding waters.

Cobalt concentrations in hydrothermal vents have previously been studied at TAG and can be

taken as potential mixing endmembers (James et al., 1995; Swanner et al., 2014). Suspended

particulate cobalt exhibited a dramatic maximum in the hydrothermal plume (12.5 pM pCo,

Fig. 8), a 20-fold increase over the background concentrations (~0.6 pM pCo) as observed in

the particulate cobalt ocean section (Fig. 3). The differences between the dissolved cobalt and

dissolved iron and manganese within the hydrothermal maximum were staggering: iron

concentrations reported in the plume are almost 4 orders of magnitude higher than that of

cobalt, and manganese concentrations are 3 orders of magnitude higher (Hatta et al., 2015).

These iron and manganese features were observed at adjacent stations as well, while the

cobalt feature was confined to the near-ridge region at USGT11-16. These large differences

should be considered in light of their respective background concentrations away from the

vents: Cobalt concentrations are approximately 1 order of magnitude less than that of iron,

and approximately 4-fold less than that of manganese. Thus, even taking the higher relative

concentrations of iron and manganese into account, the near-field net hydrothermal source

difference between metals was major. Dramatically high concentrations of particulate iron

oxyhydroxides were also observed at the vent site (~50nM pFe, Ohnemus and Lam, 2015),

which likely controlled the overall modest increase in dissolved cobalt distributions by

dominating cobalt scavenging. Evidence for localized scavenging at the vents was also





observed in negative dCo:P relationships in samples closest to the TAG vent site (see
companion manuscript, Saito et al., submitted, their Figs. 6 and 7, bottom depths of station
1116). Interestingly, no similarly dramatic increase in pMn was observed, implying the near-
field Co scavenging was related to iron oxide precipitation rather than Mn-oxidizing bacterial
activity (Ohnemus and Lam, 2015).
Cobalt comprises a much smaller fraction of crustal material than these other metals, so
hydrothermally leached crust may be expected to reflect similar ratios to that found in crustal
material. When dissolved cobalt and iron reported above are normalized to manganese, the
relative values are consistent with the dilution of hydrothermal fluids measured at Rainbow,
one of the vents located at TAG (Fig. 8, (Hatta et al., 2015; James et al., 1995)). This
observation is also consistent with the relative concentrations detected just above the Mid-
Atlantic Ridge in the South Atlantic where no notable cobalt feature was found amidst
pronounced Mn and Fe plumes (Fig. 8, (Noble et al., 2012; Saito et al., 2013)), likely due to
the dilution of the hydrothermal cobalt contributions to below the NADW background
inventory. The observation of the hydrothermal cobalt signal in the North Atlantic zonal
transect but not the South Atlantic zonal transect was likely a result of the targeted and close
sampling of a known hydrothermal field, while the South Atlantic transect accidentally
observed a large hydrothermal plume without prior knowledge of any nearby potential
hydrothermal field sources to the sampling locations. Circulation patterns at vent sites may
also be characterized by circulation patterns that are constrained by the bathymetry of the
spreading system, creating a local swirling effect that may allow particle reactive metals to
precipitate with less lateral advection (Baker et al. 1995).  We also observe an interesting
trend among these ratios. Where the ratio of Co:Mn increases with dilution (*i.e.* from direct
hydrothermal fluid sampling, to intentional plume sampling around a temperature anomaly, to
accidental plume sampling with no observed temperature anomaly), the Fe:Mn ratio follows
the opposite trend. Vents from different areas are known to be characterized by different
source ratios of metals so this comparison may be coincidental. It is also consistent with what
would be expected of the relative differences in biogeochemical oxidative removal rates of
these metals, where cobalt tends to be scavenged more slowly than manganese and iron.
Additionally, it might also be explained by a difference in the relative importance of
hydrothermal vs. other sources of dCo and dMn to their respective deep ocean inventories.



### 3.4.6 **No discernible source of cobalt from Mediterranean Outflow Water**

Off the coast of Portugal near the opening to the Mediterranean Sea, Mediterranean Outflow Water (MOW) was sampled at Stations USGT10-01 to USGT10-08 within the Meridional section at a depth range of ~1000-1500m (Jenkins et al., 2015)(Figs 2 and 3). There was no discernible MOW source of elevated cobalt to other Atlantic water masses, similar to a lack of dissolved iron in MOW observed on this transect (Hatta et al, 2015). Previous studies have observed slightly elevated concentrations of these metals (Bowie et al., 2002; Morley et al., 1997), and high aluminum concentrations associated with MOW have been used to help trace the presence of MOW (Measures et al., 2015; Measures et al., 1995). This section also revealed slight increases in Pb concentration coincident with MOW (Noble et al. 2015). Significantly elevated lithogenic particle loads were also identified in these samples (Ohnemus and Lam, 2015) though, suggesting perhaps that any labile particulate cobalt may have already been released or otherwise removed from these particles. Cobalt concentrations were quite uniform within MOW during this expedition, which may be reflective of the short residence time of the Mediterranean Sea of ~100 years (Lacombe et al., 1981).

### 3.5 **Variable Influence of Deep Margin Nepheloid Layers**

The role of the sediment-water interface and sediment resuspension in nepheloid layers has long been thought to influence the distributions of trace metals, yet few expeditions have sampled these deep regions, and none we are aware of for dissolved cobalt. The North Atlantic zonal GEOTRACES section provided a useful opportunity to examine these potential interactions, and evidence suggests that the differences in cobalt concentration in deep waters along the margins are more complex than dissolution and release near the bottom: the processes that stir up and create nepheloid layers, which provide increased surface areas for scavenging removal of cobalt, can also promote benthic release of dissolved cobalt. Here we observed a variable influence of nepheloid layers on the distributions of dissolved cobalt (not contributing or removing cobalt), that may be related to particle density and composition.

Along the western margin of the North Atlantic transect, a pronounced nepheloid layer was sampled at USGT11-06 and USGT11-08 (characterized by suspended particulate mass maxima (SPM) of 763 µg/L at USGT11-08, USGT11-06 was not sampled for particles), and a less dramatic but larger nepheloid layer at USGT11-10 (SPM maximum of 40 µg/L, Lam et




al. 2015, Fig. 9). The bottom two depths at USGT11-06 (~4500 – 4900m) showed  slightly
(~3 pM) elevated total dissolved cobalt relative to the waters above. This could be due to
contributions from different water masses that have experienced differing degrees of
scavenging over time and/or release of cobalt from the nepheloid layer sampled therein. The
bottom two samples are associated with a higher percentage influence of Iceland Scotland
Overflow Water (ISOW) relative to the four shallower samples above (~3500 -4300m), which
correspond with a higher percentage of Denmark Straight Overflow Water (DSOW, (Jenkins
et al., 2015)). These water mass differences are supported by changes in the $^{206}Pb/^{207}Pb$
isotope signature as well, which gives some indication of the contributions from different
water masses due to the isotope signatures captured at their respective outcroppings, both
spatially and temporally (Noble et al. 2015). At the BATS crossover station (USGT11-10),
several samples were collected near the bottom, and again, a slight increase in labile and total
cobalt was observed. A thick nepheloid layer extended several hundreds of meters into the
water column but produced a gradual and small decrease in transmittance voltage, unlike the
dramatic decrease observed at USGT11-08.
It should be noted that transmittance voltage is not the best indicator of suspended particulate
mass due to differential sensitivity to particles of different composition. It does give an
indication of the presence of these features, which have been confirmed and characterized
more quantitatively by chemical determinations of SPM (Lam et al. 2015). The deepest
samples show a slight increase in both the total (8pM) and labile (5pM) concentration
approaching the deepest sample, possibly suggestive of some resuspended and redissolved
particulate cobalt. This slight increase in dissolved cobalt concentration was accompanied by
a strong increase in suspended particulate cobalt (Fig. 9). If this increase were due solely to
dissolution of resuspended particulate cobalt, a much stronger dissolved cobalt increase may
have been expected at USGT11-08. At USGT11-08, however, where the most dramatic
decrease in transmittance voltage was observed, and the highest deepwater concentration of
suspended particulate cobalt was observed, there was no significant increase or decrease in
dissolved total or labile cobalt. No major differences were observed among the water masses
occupying the bottom depths between USGT11-06, USGT11-08, and USGT11-10.  These
contrary observations suggest a balancing act between the source function of resuspension
(benthic release of dCo) and the sink function of the nepheloid layer itself (increased surface
area for scavenging, i.e. SPM).  It is possible that at USGT11-10, the SPM maximum (40
μg/L) was not high enough to overcome the source from benthic release.  At USGT11-08;



however, the much higher SPM (763 µg/L) may have sufficiently scavenged any cobalt from
benthic sources. The composition of the lithogenic particles that dominate these nepheloid
layers, are also not particularly good scavengers (Lam et al. 2015), and this composition is
likely a key factor driving the differences observed between the wester and eastern margins.

5       Along the eastern margin, USGT10-09 also sampled a notable, though smaller,

nepheloid layer (SPM max of 44 µg/L, Lam et al. 2015), and in this case, a distinct minimum
was observed in both total dissolved cobalt and labile cobalt, coincident with a significant
maximum in suspended particulate cobalt (Fig. 9). This nepheloid layer, unlike the western
margin nepheloid layer, contained particulate manganese and iron oxyhydroxides, which are
likely much more efficient scavengers of cobalt than lithogenic particles (Lam et al. 2015).
Labile cobalt was undetectable in the deepest sample, suggesting that all cobalt was tightly
complexed or that the labile fraction had been scavenged away. This decrease in labile cobalt
was also observed at Stations UGST10-10 and USGT10-11, where much smaller nepheloid
layers were observed, coincident with relatively small concentrations of resuspended
particulate cobalt (Fig. 9). This dramatic difference where some bottom samples along the
western margin show slight enrichment while some bottom samples along the eastern margin
show strong depletion, both in the presence of notable nepheloid layers and elevated
particulate cobalt was intriguing (e.g. compare USGT11-10 to USGT10-09), and
demonstrates the chemical diversity of the dissolved and particulate phase interactions. Future
characterization of the particle composition in these margin samples and process studies could
provide mechanistic explanations for the capacity of particles acting as sources or sinks of
cobalt to the dissolved pool.

### 3.6 Inverse Relationship with Oxygen and Implications for Deoxygenation

In intermediate depths, and particularly within the eastern margin cobalt plume, cobalt and
dissolved oxygen showed a significant inverse relationship (Fig. 10). As mentioned in
Section 3.4.1, elevated concentrations here are likely driven by a combination of a
sedimentary source involving reductive dissolution and advection, as well as remineralization
of sinking biological material. Both of these processes are linked to oxygen in an inverse
fashion and are the likely explanation for this linear relationship. The low $O_2$ concentrations
also allow for the persistence of high dissolved cobalt through slowed oxidation into
manganese oxide particles (Moffett and Ho, 1996). In our previous study of the South





Atlantic, we discussed the potential for trace metal ocean inventories to increase as a result of
ocean deoxygenation (Noble et al., 2012), based on observations compiled by Stramma et al.
of deoxygenation within the major oxygen minimum zones across the world oceans over the
last 50 years (Stramma et al., 2008). In our South Atlantic work, we made a back-of-the-
envelope calculation of the influence of increasing ocean deoxygenation and potential for
increasing sedimentary release of cobalt assuming a linear relationship in concert with
deoxygenation rates determined by Stramma et al. While this assumption that the linear
inverse $dCo:O_2$ relationship would be constant moving forward in time is simplistic since its
mechanistic basis remains unknown, it provides a useful first approximation of potential
increases in cobalt ocean inventories.
Here, we apply the same approach to the North Atlantic OMZ to estimate the potential
increase in cobalt inventory in the upper 1000m of the North Atlantic that may be attributed to
deoxygenation. Within the low oxygen region of the North Atlantic, between 300-800m
depth, cobalt and $O_2$ display an inverse relationship with a slope of -0.56 pmol dCo $\mu mol^{-1}$ $O_2$
(n = 73, $r^2 = 0.89$), very similar to our findings in the South Atlantic (-0.56, $r^2 = 0.73$, (Noble
et al., 2012)).  This suggests that a similar chemistry governs the relationship, not an aspect of
a preserved coastal $dCo:O_2$ signature, and could be related to the oxygen needs of manganese
oxidation and co-oxidation.  Stramma et al. estimated an ocean deoxygenation rate for the
North Atlantic OMZ of -0.34 $\mu mol$ $O_2$ $kg^{-1}$ $y^{-1}$ over the past 50 yr (Stramma et al., 2008).
Together, these relationships can be used to estimate potential future increases in cobalt
concentrations within the oxygen minimum zone. An upper 1000m cobalt inventory from
USGT11-10 (BATS) across the basin to the most coastal station at USGT10-09 was estimated
by summing the estimated dissolved cobalt within each of many trapezoid shaped water
parcels, utilizing each depth and the distance between adjacent stations to interpolate cobalt
concentrations between stations and between samples. We then estimated the potential impact
of deoxygenation rates on the cobalt inventory within the OMZ, and that subsequent impact
on the upper 1000m inventory as a whole by using the $dCo:O_2$ relationship and North Atlantic
deoxygenation rate described above. The upper 1000m is utilized because of the low $O_2$
waters found in the 250-850m depth range, although this calculation could be easily modified
for other depth ranges. Extrapolating forward 100 years, using this simple calculation we
estimate that the cobalt inventory in the upper 1000m of the North Atlantic could increase by
20% in the next 100 years. These large potential changes in upper ocean inventories may have
implications for the ecological balance within this basin. This increase is also two-fold higher



than the ~10% estimated for the South Atlantic, which is largely due to the two-fold higher
rate of deoxygenation reported for the North Atlantic OMZ (-0.34 μmol $O_2$ kg$^{-1}$ y$^{-1}$) relative to
the South Atlantic OMZ (-0.17 μmol $O_2$ kg$^{-1}$ y$^{-1}$, (Stramma et al., 2008)). These results imply
a need to consider the influence of changing oceanic oxygen on the biogeochemistries of
metals and their influence on marine ecology.

### 3.7 Relationships of dissolved and labile cobalt with soluble reactive phosphate in the upper Atlantic Ocean

Dissolved cobalt distributions in the oceanic upper water column are influenced by biological
processes such as uptake and remineralization (Noble et al., 2008). The nutrient stoichiometry
of the aggregate microbial ecosystem can be inferred using a similar approach to that
originally used by Alfred Redfield for dissolved and particulate nitrogen and phosphate
(Redfield et al., 1963), where linear relationships between dissolved cobalt and soluble
reactive phosphate can be interpreted as time-integrated signals of the extent of cobalt
utilization by the resident phytoplankton community and their subsequent remineralization
from the biological particulate phase. The aggregate slope of this correlation is termed the
"ecological stoichiometry" for their inferred biological usage (Sterner and Elser, 2002). An
emerging distinguishing feature of cobalt relative to other macro (N and P) and micronutrients
(Zn and Cd) is a much larger range in stoichiometries when different oceanic regions are
compared (Noble et al., 2012; Noble et al., 2008; Saito et al., 2010) (Baars and Croot, 2015;
Bown et al., 2011; Sunda and Huntsman, 1995). The production of large GEOTRACES
datasets provides an opportunity to explore this variability in stoichiometry and the processes
behind them. Here, we describe broad regional differences in the Co:P relationships in the
North Atlantic. A detailed and finer-scale analysis of these relationships and their ecological
interpretations are discussed in a companion manuscript (Saito et al. in prep).
When the North and South Atlantic zonal datasets (NAZT and CoFeMUG) were compared in
an aggregate scatter plot (Fig. 10), there were two notable differences. First, there is a shift
toward lower phosphate concentrations relative to cobalt concentrations in the North Atlantic
when compared to the South Atlantic. This is likely due to the lower surface phosphate
inventory observed in the North Atlantic relative to the South Atlantic (Noble et al., 2012; Wu
et al., 2000), as was evident in comparisons of nitrate+nitrite versus phosphate on these two
transects and the higher phosphate axis intercept in the South Atlantic (Fig 10A). Second,
there was also an offset in cobalt abundances, with higher total dissolved cobalt in the North



Atlantic that consistently approached ~150pM in the eastern North Atlantic (Fig. 2), likely
due to the higher atmospheric cobalt flux and resultant concentrations in the North Atlantic
relative to the South Atlantic. As mentioned earlier, the North Atlantic experiences significant
aeolian input from the Saharan Desert compared to the much lower dust inputs to the South
Atlantic (Noble et al., 2012), and aeolian deposition is not considered to be a major source of
phosphorus. This offset can also be seen in the dCo:$O_2$ plot as a vertical shift (Fig. 10), also
likely caused by the higher dust contribution in the North Atlantic and an overall greater
inventory.
Dissolved labile cobalt (LCo) also showed linear relationships with phosphate in the North
Atlantic (Fig. 11), where labile cobalt is defined as the sum of the free cobalt and the cobalt
bound to weak ligands (Saito et al., 2004). To our knowledge, this is the first report of linear
relationships between labile cobalt and phosphate. Unlike the frequent observations of excess
strong iron binding ligands in oceanic photic zones (Buck, 2007; Buck et al., 2015; Rue and
Bruland, 1997), strong cobalt binding ligand concentrations tend to be less than or equal to
total Co, allowing frequent detection of labile cobalt in the water column, and the potential for
large swings in bioavailability of cobalt. The coherent lower slope of the LCo:P relative to
total cobalt in the water column below the upper photic zone is particularly intriguing as it
contrasts an apparent steeper and less coherent LCo:P  in the upper photic zone. The low
labile cobalt in the upper photic zone was expected due to phytoplankton and microbial
uptake, reflective of the scarcity of this labile cobalt form and resulting in it comprising a
small fraction of total dissolved cobalt there. The correlation labile cobalt with phosphate in
the ocean interior (Fig. 11) implies a remineralization source from decaying phytoplankton
material. It is possible that cobalt taken up by phytoplankton and prokaryotic microbes for use
in enzymes or vitamin $B_{12}$ is present in a proteinacious form intracellularly that is susceptible
to degradation with proteolytic activity upon sinking and results in its release as labile cobalt.
This would be in contrast to strongly complexed cobalt that is formed through insertion into
corrin rings of the $B_{12}$ precursor by cobaltochelatase enzymes (Bonnet et al., 2010; Rodionov
et al., 2003; Saito et al., 2005). This duality in cobalt's chemical forms, having both
complexed and labile forms, adds a layer of variability in availability and geochemical
cycling that is similar to that of dissolved and colloidal size fractions of iron (Bergquist et al.,
2007; Fitzsimmons and Boyle, 2014). This fraction of the labile cobalt is likely a less
protected inventory relative to scavenging processes and even though it is a small component
of the dissolved cobalt inventory, it could play a major role in cobalt  biogeochemical cycling.





### 3.7.1 Variation in the depth of the cobaltclines: evidence for dynamic biogeochemistry

Examination of individual vertical profiles, with a focus on several stations from USGT10, further reveals the dynamic nature of cobalt chemical speciation and its influence on cobalt biogeochemical cycling in the photic zone of the North Atlantic (Fig. 12). With the exception of the labile cobalt at station USGT10-06, dissolved cobalt, labile cobalt and phosphate are all drawn down to their lowest concentrations in the mixed layer. By comparing these species to biological and physical proxies such as fluorescence and density, a few subtle differences emerge that are influenced by changes in the mixed layer depth and chlorophyll max. At all 4 stations, the gradient in total cobalt concentrations, or the "total cobalt-cline" coincides with the base of the mixed layer. At USGT10-09, the "phosphocline" and the "labile cobalt-cline" also coincide with the base of the mixed layer. Here, the waters are particularly productive as seen by the intensity of the fluorescence peak, and the three analytes reach relatively high concentrations due to upwelling, potential aerosol inputs (3.4.4), and sedimentary sources from the plume as discussed earlier (3.4.1). The chlorophyll maximum was shallow and pressed up against the base of the mixed layer as a result of the upwelling.

We can compare this to the patterns observed at USGT10-03, where less productivity was observed due to a lack of significant external nutrient sources and correspondingly smaller local sub-mixed layer nutrient inventories (note the scale difference). Here, the total cobaltcline again coincides with the mixed layer depth (77 m), but phosphate and labile cobalt are both drawn down below detection, much deeper, into the middle of the chlorophyll maximum (99m, Fig. 12, (Shelley et al., 2012a)). In the other two stations as well, the chlorophyll maximum is smaller and deeper, and the labile cobaltcline follows the phosphocline, where the total cobaltcline remains coincident with the mixed layer depth, revealing a confluence of processes that are occurring on relatively short timescales. With a short residence time of 0.32 y in the upper water column (upper 100m; Saito and Moffett, 2002), the tug of war between biochemical and geochemical processes within one profile can be seen. This offset between the labile and total cobaltclines suggests that biological processes act quickly enough to complex labile cobalt that enters the chlorophyll maximum at a rate faster than upward mixing. While pigment samples from this transect were lost during freezer failure, making assessment of the biological contributions to these variations difficult, we know that generally the coastal regions of the Atlantic have more eukaryotic phytoplankton




representation, while the oceanic regions are dominated by picocyanobacteria and
picoeukaryotes (Olson et al., 1990), and that picocyanobacteria are sources of metal binding
ligands in both open ocean and coastal waters, including for cobalt and copper (Moffett and
Brand, 1997; Saito et al., 2005). In the Mauritanian Upwelling, the inventories are higher and
productivity is more intense, but the chlorophyll maximum is pressed up against the mixed
layer so the differences in rates of uptake, complexation, diffusion, mixing, and upwelling
cannot be easily separated.

### 3.7.2 Loss of cobalt from intermediate and deep waters by scavenging

In addition to the variety of sources that contribute cobalt to the North Atlantic
described above, there is evidence for a dissolved sink from the pelagic water column
throughout this North Atlantic zonal transect. This is evident in the vertical structure of
profiles that, unlike nutrient-like elements such as phosphate and zinc, decrease precipitously
at intermediate depths: below ~1000m in the western and northeastern Atlantic profiles and
below ~600m on the eastern profiles off of Mauritania (Fig. 2). These changes in vertical
structure likely reflect a shift in the balance between long-term scavenging removal processes
occurring on horizontally advecting water masses relative to the vertical input of dissolved
cobalt from remineralizing sinking particles. These scavenging processes can be observed in
aggregate through an examination of the relationship between total dissolved cobalt and
soluble reactive phosphorus (dCo:P) across the basin that displayed a downward curl,
reflective of a loss of total dissolved cobalt relative to phosphate consistent with a preferential
scavenging of total dissolved cobalt (Fig. 13), likely into bacterially-formed manganese oxide
particles (Lee and Tebo, 1994; Moffett and Ho, 1996). Examination of the water masses
calculated through OMPA analysis (Jenkins et al., 2015) associated with datapoints in dCo:P
space showed water masses with unique signatures. In particular, the combined deep DSOW-
AABW-ISOW (~ >3000m depth, Denmark Strait Overflow, Antarctic Bottom Water, and
Iceland-Scotland Overflow) and CLSW (~2000-3000m, Classical Labrador Seawater) water
masses were the major contributors to the North Atlantic scavenged "curl" feature (Fig 13C)
implying loss of cobalt relative to phosphate in those water masses during their long-term
advection. This cobalt curl feature is also evident in the South Atlantic zonal section as well,
largely overlapping with the features observed here (Fig. 13A and 13B). Finally the amount of
dissolved cobalt at intermediate depths decreases from the North Atlantic to the South



Atlantic, consistent with a scavenging loss with thermohaline circulation. The accompanying
manuscript (Saito et al. submitted) conducts further statistical analysis and discussion of the
scavenging process through a profile-by-profile examination of the dCo:P relationship.
Notably there were also instances where regional circulation influences the otherwise
generally "typical" hybrid-type profile structure. For example, vertical structure was notably
perturbed at station USGT10-07 where sharp concentration gradients appear coincident with
jetting intrusions of water masses as indicated by oxygen concentration, water mass analysis
(Jenkins et al., 2015), and $SF_6$ tracer age (Smethie et al. in prep) (Fig. 14).
## 4 Conclusions
The dissolved and labile cobalt datasets for the North Atlantic zonal transect reveal numerous
sources of cobalt to the North Atlantic. A large plume of cobalt was observed at ~400 m depth
within the Mauritanian Upwelling along the eastern margin, which reflect eastern margin
dissolved cobalt sources that are concentrated in the OMZ by phytoplankton uptake, export
and remineralization and dust inputs. The western margin also displayed an elevated cobalt
feature at intermediate depths characterized by ULSW, likely due to the mobilization of
cobalt from continental shelf sediments either before or after subduction of the ULSW
watermass. Hydrothermal and aeolian sources were detectable but small relative to these
larger ocean features. Variable sources and sinks of cobalt were observed in deep margin
nepheloid layers, suggesting that particle composition and sediment redox gradients may play
an important role and should be taken into consideration in future studies. Using
deoxygenation rates and a relationship between cobalt and $O_2$, we estimate that the cobalt
inventory in the upper 1000m of the North Atlantic may increase by 20% in the next 100
years due to ocean deoxygenation, approximately twice that previously estimated for the
South Atlantic OMZ region. Differences in the ecological stoichiometry of cobalt observed in
the upper water column imply that a wide variety of cobalt utilization regimes exist.  The
processes of uptake and remineralization exerted control on cobalt in the oligotrophic surface
waters of the North Atlantic, demonstrated by correlations with phosphate, and the strong
drawdown of phosphate, nitrate, and total dissolved and labile cobalt.  When low salinity,
coastal, metal inputs and physical processes imposed a strong influence along Line-W, these
correlations were obscured, muting the influence of biological processes that operate to
couple these species. Combining growing datasets of cobalt coming from the GEOTRACES



program with future biochemical studies will improve our understanding of the influence of
cobalt biogeochemical cycling and its interaction with ocean marine ecology.
Increasing anthropogenic cobalt use due to growth in the economic market for cobalt in
lithium batteries and other sources poses the potential to drastically change global oceanic
cobalt distributions since the potential environmental impact of cobalt pollution is currently
unknown. As such, it is more important than ever to establish a baseline understanding of
cobalt distributions in the ocean to provide important insight into its oceanic biogeochemical
cycling and to inform potential future impact of industrial use of cobalt on the ocean
inventory.

**Author contribution**

At-sea and laboratory analyses of total dissolved and labile cobalt were conducted by A.
Noble. Data analysis and manuscript writing were conducted by A. Noble, M. Saito, and N.
Hawco.   Particulate sample collection, analyses, and interpretations in the text were
conducted by P.J. Lam and D. Ohnemus.

**Acknowledgements**

We would like to thank the GEOTRACES Expedition Team, including the Chief Scientists
Ed Boyle, Bill Jenkins, and Greg Cutter, and the GEOTRACES sampling team.   We also
thank the Captain and Crew of the R/V *Knorr* for their outstanding support of science. We
also gratefully acknowledge support of funding agencies on the following grants: the US
National Science Foundation (NSF-OCE 0928414, 1233261, 1435056) and the Gordon Betty
Moore Foundation (Grant 3738).



**Figure Captions**
**Figure 1.** Map of USGT10 and USGT11 expedition tracks.
**Figure 2.** Dissolved profiles of total and labile cobalt for USGT10 and USGT11. Several
stations of note are discussed more fully in the text, but some of note include USGT11-10
(BATS), USGT11-16 (TAG), USGT10-09 (Station closest to the Mauritanian coast),
USGT11-1 to USGT11-08 (Stations along Line-W). USGT10 was sampled during the fall of
2010, and USGT11 was sampled during the fall of 2011.
**Figure 3.** (A) Full depth section of total cobalt, (B) full depth section of labile cobalt, and (C)
full depth section of particulate cobalt with the meridional section in the right panels and the
zonal section in left panels. These ocean sections were created using Ocean Data View. The
dissolved sections were created using VG gridding and extrapolated lengths of less than 70
permille in either y or x direction for any of the section representations.
**Figure 4.** Examination of storage effects and use of gas absorbing satchels for preservation on
a vertical profile from Station USGT10-9 collected near Mauritania. "Preserved" samples
were kept refrigerated in heat-sealed bags with gas absorbing satchels, while "4 months"
samples were only kept refrigerated. The preserved samples showed excellent recovery after
four months in cold storage compared with at-sea measurements, while non-preserved
samples showed significant loss of dissolved cobalt.
**Figure 5.** Section of dissolved cobalt and dissolved silicate along the western margin of the
North Atlantic (Line-W). Upper Labrador Sea Water, identified generally between 700 and
1500 m depth by OMPA analysis (Jenkins et al., 2015) carries elevated concentrations of
cobalt and is depleted in silicate. Elevated cobalt concentrations may be due to interaction of
this water mass with sediments along the wide coastal shelf.
**Figure 6.** Total and labile cobalt show strong correlations with phosphate in the North
Atlantic Subtropical Gyre, but this relationship is not observed along Line-W. Along Line-W,
strong relationships between (A, B) salinity and total cobalt and (A, C) salinity and labile
cobalt were observed in surface waters.
**Figure 7.** Surface transects for (A-B) the north Atlantic zonal section USGT10 and USGT11
(top panels), and (C) the South Atlantic zonal section CoFeMUG (Noble et al., 2012). (D)
Relationships between dCo:dAl and correlations for northeast Atlantic stations USGT10-08 to
USGT10-12. (E-F) Comparison of vertical profiles of dCo between the North and South
Atlantic zonal sections near the African Coast. Distances of stations to the African coastline





were ~530km for Tenatso (USGT11-24) and 210 km from USGT10-09 in the North Atlantic,
and 1300 km for Station 13 and 790 km for Station 15 from the CoFeMUG Expedition in the
South Atlantic.
**Figure 8.** (A) Manganese-normalized cobalt and iron concentrations in hydrothermal fluid
and above the Mid-Atlantic Ridge. Hydrothermally leached crust may reflect ratios of
reducible metals that are similar to that found in crustal material. Normalized cobalt and iron
values are consistent with what might be expected by dilution of hydrothermal fluids
measured at TAG. This is consistent with dissolved data from above the Mid-Atlantic Ridge
at 9 ° S as well. The ratio of Co:Mn also increases with presumed dilution (from vent fluids
to TAG to 9 ° S), while the Fe:Mn ratio decreases. This is consistent with the expected
relative oxidative removal rates of these metals: Co < Mn < Fe. (B) Profiles of dissolved total
and labile cobalt above the TAG hydrothermal vent and above the Mid-Atlantic Ridge at 9 °
S. A slight maximum in total and labile cobalt suggests that hydrothermally released cobalt at
TAG is primarily labile and that the vent may provide a small, local source of cobalt. (C)
Particulate cobalt profile at TAG. The dramatic signal of pCo is notable against the
background concentrations, but small relative to those observed for Fe and Mn (see text).
**Figure 9.** Along the margins, nepheloid layers have differing effects on the dissolved cobalt
concentration and composition, suggesting that more complex processes than simple
dissolution from resuspended particles. Thick and large nepheloid layers along the western
margin (three profiles to the left) appear to have small or insignificant effects on the dissolved
and labile cobalt profiles, while a moderate nepheloid layer along the eastern margin at
USGT10-09 (profile furthest to the right) appears to have a strong scavenging effect on
the dissolved and particularly the labile cobalt. Labile cobalt in the deepest samples here
were drawn down below the detection limit. The dramatic differences in the effect of
nepheloid layers on the dissolved and labile cobalt concentrations demonstrates great
chemical diversity in the dissolved-particulate phase interactions and suggests that
these interactions cannot be generalized by phase alone.
**Figure 10.** Aggregate nutrient stoichiometries between the North Atlantic and the South
Atlantic studies. Basin offsets were observed in comparisons between (A) nitrate and
phosphate (linear regression for South Atlantic with a slope of 17.4, $r^2$ of 0.995) and (B)
cobalt and phosphate concentrations in the upper water column of the North and South
Atlantic zonal sections (North Atlantic stations, 2-400 m, linear regression depths include 54-



300 m, with a slope of 61.4, $r^2$ of 0.90; S. Atlantic stations 1-19, 54-300m, same depths for
linear regression, slope of 52.6, and $r^2$ of 0.83). (C) Different slopes and intercepts are
observed across many regions of the world oceans that have been studied in the literature
(Martin et al., 1989; Noble et al., 2012; Saito et al., 2010). These differences arise from
variability in cobalt and phosphate utilization and supply. (D) Linear relationships occur
between cobalt and oxygen for USGT10, USGT11, and for previous work done in the South
Atlantic.
**Figure 11.** Co:P ecological stoichiometry observed across different regions in the North
Atlantic. A tighter correlation is observed where labile cobalt becomes detectable and the
correlation is observed down to differing depths depending on the strength of the processes
that affect cobalt and phosphate biogeochemical cycling. Processes that affect both species
similarly in time and space will tend to tighten the correlation and deepen the depth to which
it is observed (e.g. USGT10-01- USGT10-06). Where scavenging or reductive dissolution
may influence the two species differently, the correlation may be more diffuse or not
observed at all (CoFeMUG Sta. 8-17).
**Figure 12.** In the upper water column, labile cobalt is often drawn down above and within the
chlorophyll maximum, following the phosphocline. The total cobaltcline appears to be more
closely associated with mixed layer depth. Comparing the differences among the stations that
do and do not experience upwelling, it is apparent that biological processes are capable
of complexing labile cobalt at a rate faster than upward mixing. This is seen by
comparing USGT10-03 (located in oligotrophic waters north of the
Mauritanian Upwelling) to stations USGT10-12 (located within the Mauritanian Upwelling).
At both locations, the labile cobalt is drawn down below detection into the chlorophyll
maximum. Station USGT10-03 is located in oligotrophic waters, with a smaller deep
inventory of cobalt (~50pM at the chlorophyll max, increasing to ~60 at 200m). Station
USGT10-12, however, is located in waters that experience upwelling and have a much larger
deep inventory of cobalt (~70pM at the chlorophyll max, increasing to ~100pM at 200m).
**Figure 13.** Full depth relationships of total dissolved cobalt, phosphate, and AOU. In a
comparison with water mass analysis from the NAZT expeditions, clear populations of data
by water mass origin were observed in both (A) dCo:P and (B) dCo:AOU space. Data from
the Ross Sea is also shown as a potential endmember (Saito et al., 2010). dCo:P relationships
and the "cobalt curl" deviance from them were observed in the (C) Western and (D) Eastern





basins of the Atlantic, and were similar between the North and South Atlantic (South Atlantic
data from Noble et al., 2012).
**Figure 14.** Intermediate depth profile for (A) $O_2$, (B) labile cobalt (C) total cobalt (D) SF6
age and (E) water mass as a percent for USGT10-07. A strong linear correlation with $O_2$ is
observed (inset). A small but distinct maximum is observed between 400-600m and
demonstrates the capability of water mass features to influence the cobalt vertical profile at
these depths.



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

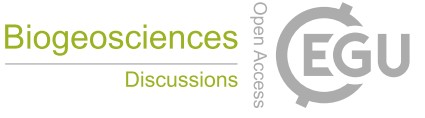

Figure 1

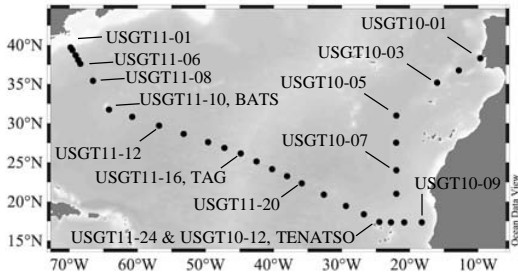

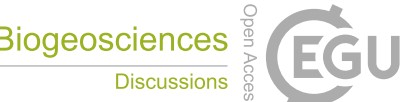

Figure 2

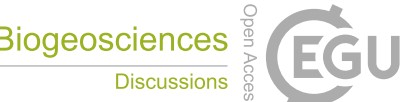

## Figure 3

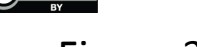

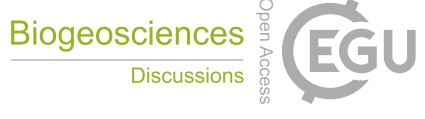

## Figure 3

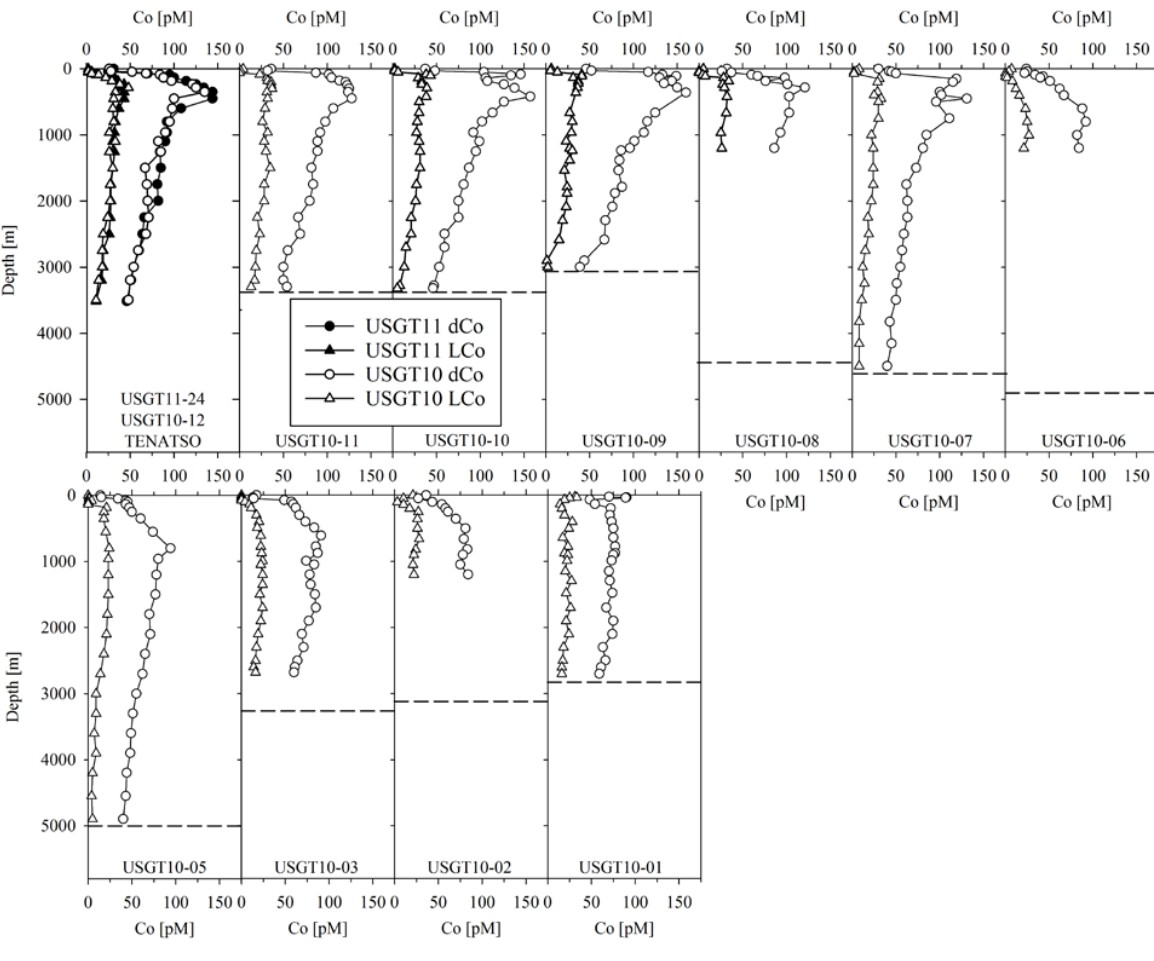



Figure 4

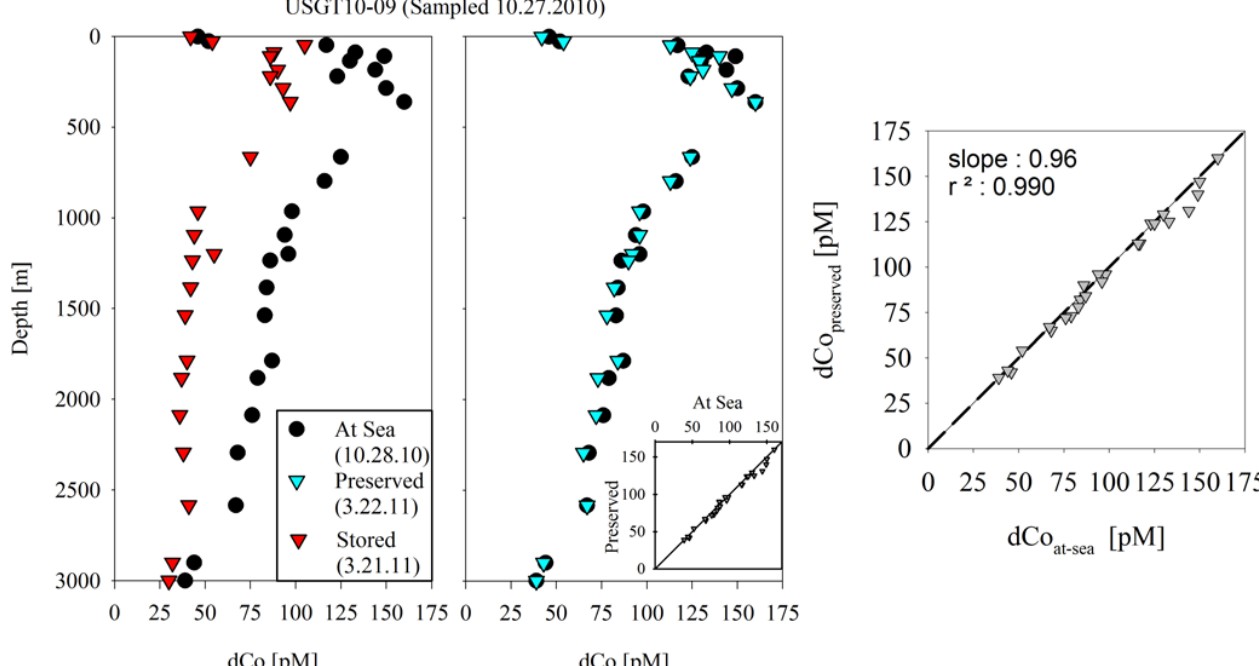




Figure 5

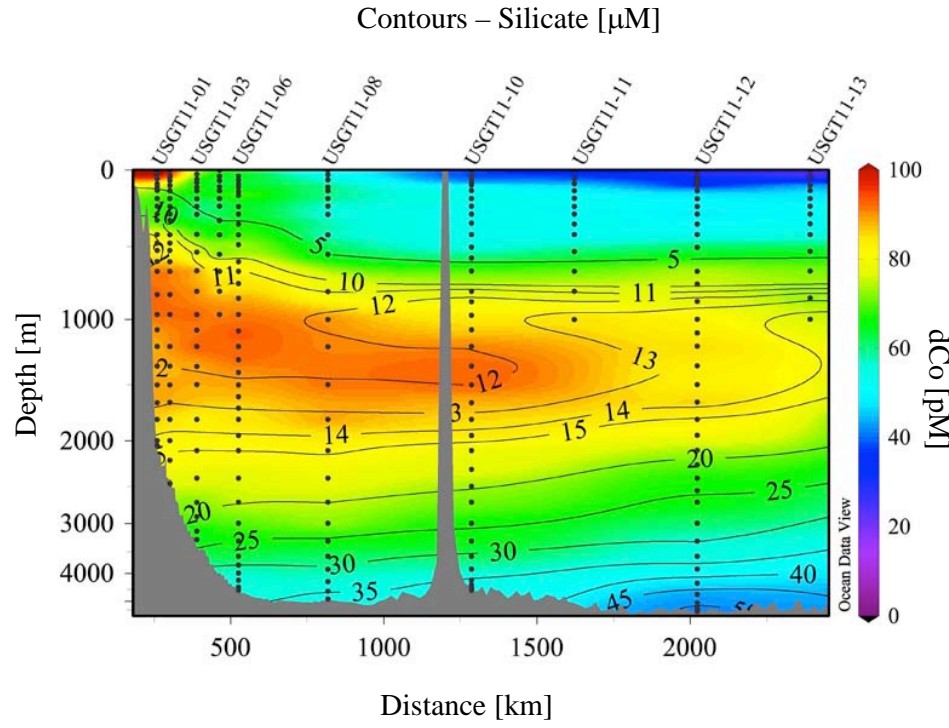



## Figure 6

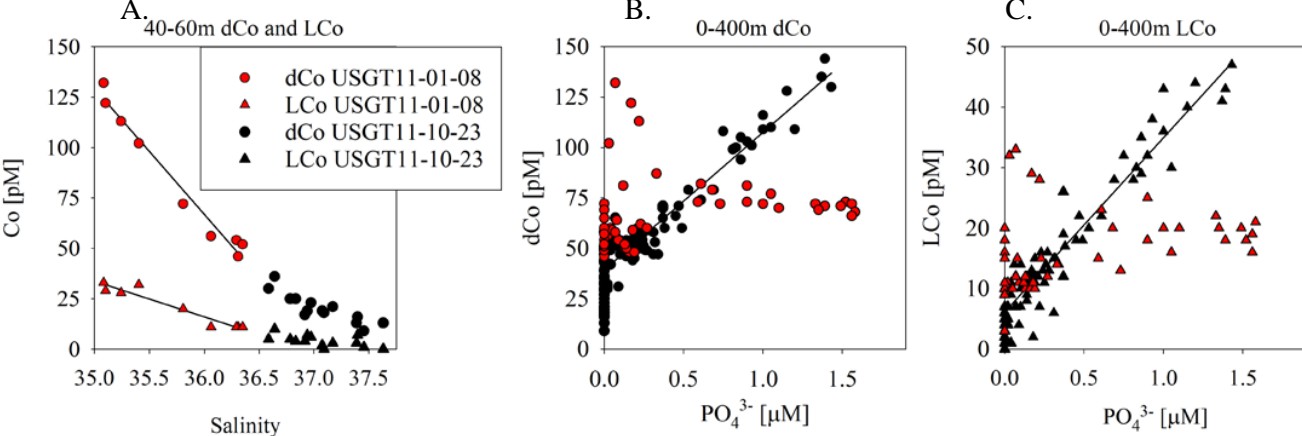



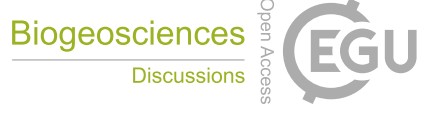

# Figure 7

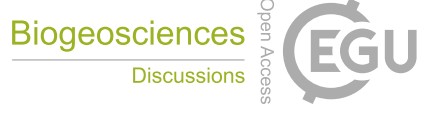



Figure 8

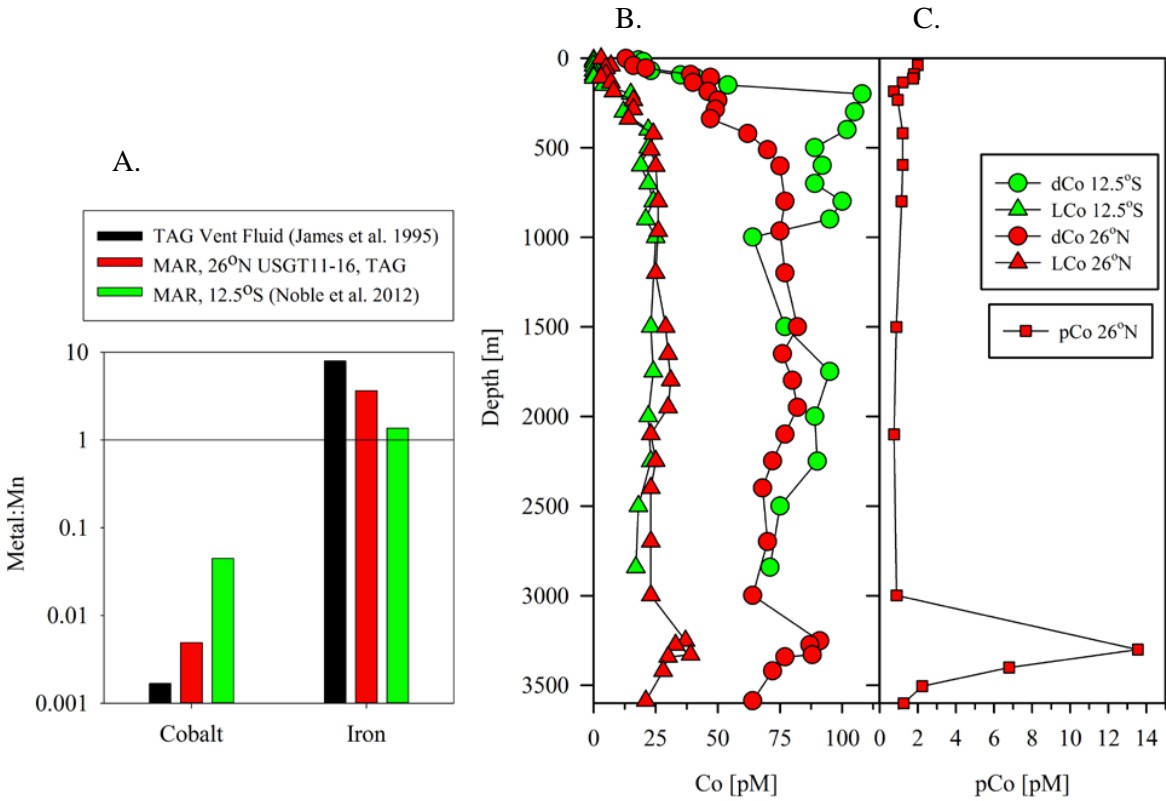



Figure 9

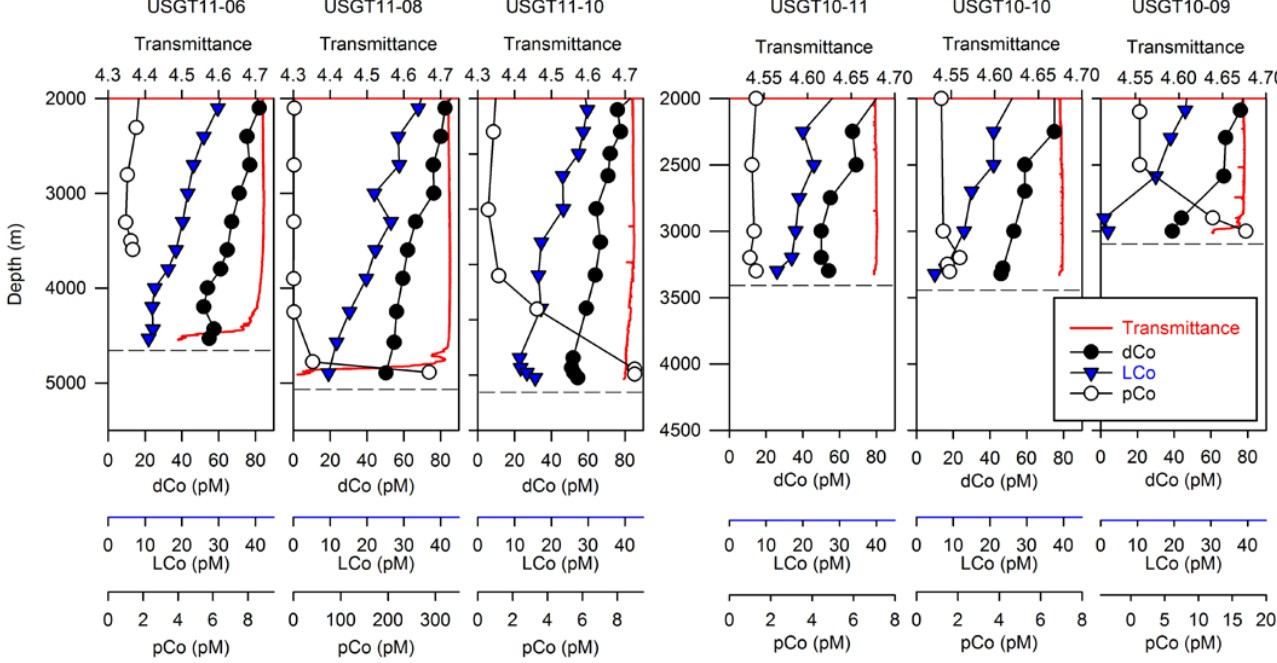





Figure 10





Figure 11

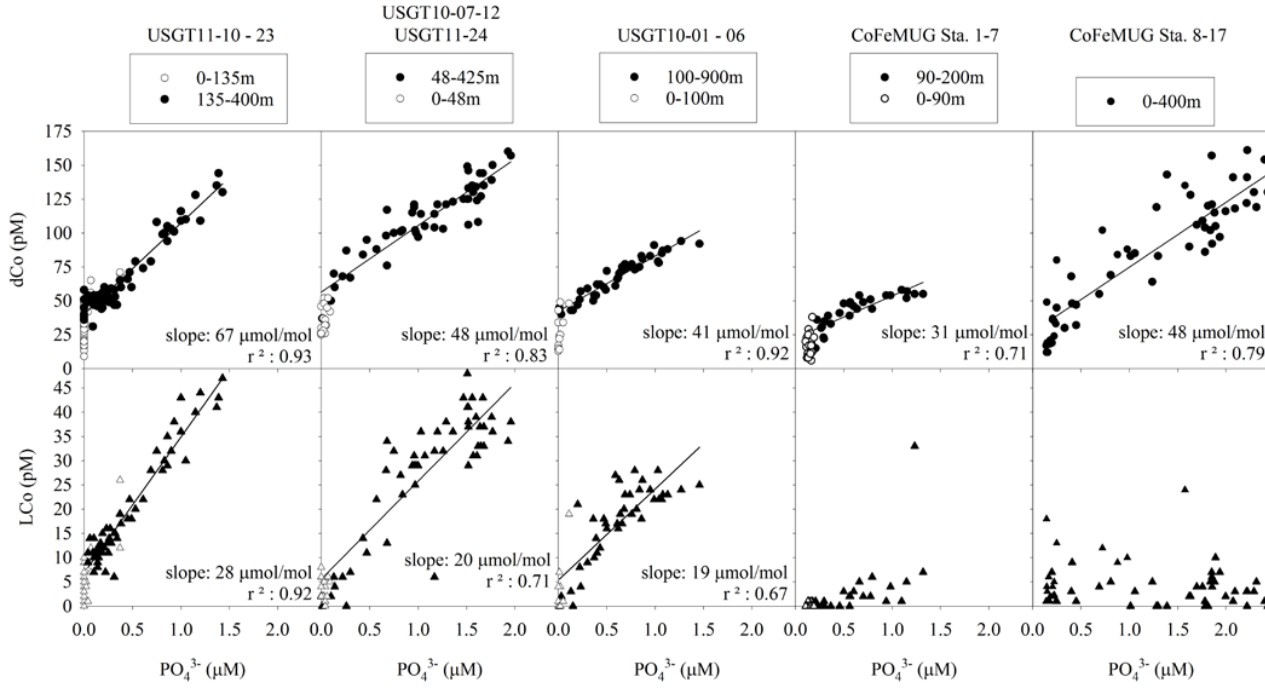





Figure 12







## Figure 13



## Figure 14

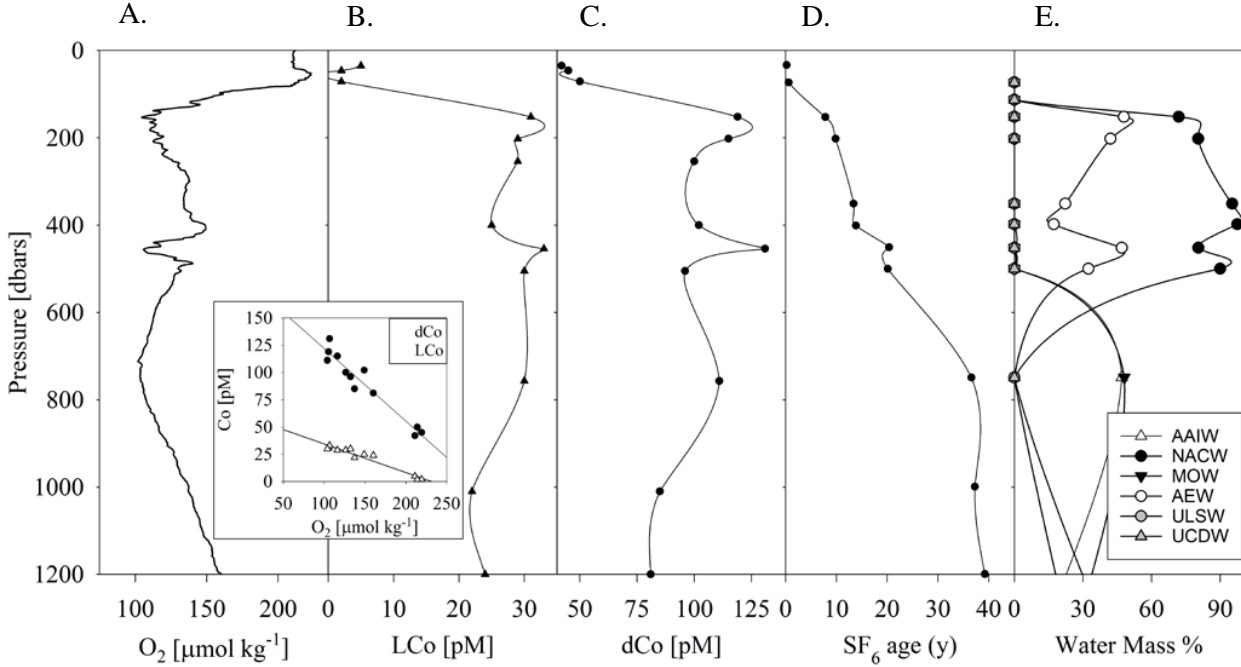