# Peer review of "Published: 1 December 2016"

_Biogeosciences, 2016_

## Referee Comment (RC1) · Anonymous Referee #1 · 6 Jan 2017

Noble et al. present cobalt measurements from two GEOTRACES transects across the North Atlantic (USGT10, Oct 2010, and USGT11 Nov-Dec 2011). Full depth sections are shown for Labile Co (LCo), total dissolved Co (DCo), and particulate Co (pCo). The study provides important methodology, a comprehensive dataset and a detailed analysis of Co sources, sinks, and cycling. The results provide significant new insight into the complex oceanic cycle of this important micro-nutrient.

The data is of high quality as evident through oceanographic consistency, intercalibration stations, and ensured by frequent blank checks and control samples. This is particularly important for the challenging electrochemical measurements for LCo and DCo at low pM concentrations. The authors also reveal an important and very interesting storage issue for Co measurements that needs to be taken into account in future studies and which may lead to new insight into Co cycling. Storage artifacts were prevented by preserving samples in heat sealed bags with O2 absorbant or immediate shipboard analysis. Missing from the methods section appears to be a detailed descripiton of pCo determination.

Where appropriate, the results from the North Atlantic are compared to an earlier study in the South Atlantic, allowing new insights resulting from differential oxygen concentrations and dust input in both regions. The manuscript is very good in separating influences of the OMZ, sedimentary processes, dust, riverine/coastal sources, hydrothermal inputs, etc. often using additional parameters (Fe, Mn, O2, aerosols, etc.) measured during the same cruise or using literature data to constrain Co cycling. Seasonality and dynamic effects are also discussed.

Overall, Noble et al., present a very careful study with a comprehensive dataset and a detailed data analysis which justifies the rather long ms. Some of the figures could be moved to the SI and the captions could be shortened. Below is a list of specific comments with suggestions for minor revisions or cosmetic changes.

Specific comments ——————— The frequently used terminology 'labile Co' and 'strong Co binding ligands' mixes kinetic (labile) and thermodynamic aspects (strong ligands). For example, in P1L16, the authors conclude that strong Co binding ligands where not in excess of total Co below the euphotic zone because labile Co was measured. Based on this and other studies, as also mentioned in the introduction, it is likely that labile Co is Co(II) and inert Co is Co(III). There is a possible conversion from labile, weakly bound Co(II) to inert, strongly bound Co(III) by oxidation. Some of the ligands that may bind Co(II) weakly may be strong ligands for Co(III) and thus it cannot be excluded that Co was remineralized as Co(II) and may, with time, be converted to inert/strongly complexed Co(III) without an apparent excess of strong Co ligands. I would suggest to add a short review of these aspects and a clarification of terminology to the speciation section in the introduction (P4L7).

P6L8 Add methodology for pCo (particulate suspended Co) determination

P9L3 A detection limit of 1.8 pM for DCo is mentioned (3x standard deviation of blank) which is significantly lower than detection limits reported in earlier studies using similar methodology. Is the peak for 1.8pM actually measurable or inferred from the intercept of standard additions? A figure showing exemplary chromatograms for the blank analysis in a new Supplemental Information document might be helpful.

P10L2 'The results demonstrate that...' -> I found this sentence complicated. Do you mean that results for GEOTRACES standards are in agreement with consensus values?

P10L16 'These results are in good agreement wiht those from the GEOTRACES intercalibration...' -> Are the results within the standard deviation of GEOTRACES consensus values? Provide a reference where these values can be found again here.

P13L28 It is stated that the storage issue for dissolved Co is more severe in the North Atlantic than in other regions. This is very interesting and the authors suggest that it may be related to dust and colloidal loading. In general, it seems almost clear that the 'loss' of Co is because Co(II) is being oxidized to an inert Co(III) form. This can be particularly pronounced in regions with high labile Co(II) and high ligand concentrations or high colloidal concentrations that can bind Co(III). Perhaps you could mention this redox aspect again in the discussion of the storage effect.

P15L29 Studies are referenced showing that the OMZs in the South Atlantic and in the North Atlantic show elevated Mn and Fe. Are the elevated concentrations in the OMZs of the two regions comparable? Does Co seem to be slower to be scavenged when reaching higher O2 than Fe and Mn, similar to your discussion of the hydrothermal Co:Mn ratios (this could be added to your conclusion of a low O2 threshold for Co plume formation, P16L17)?

P18L6 '..the relationship with salinity was similar for labile Co...' -> maybe add the

fraction of labile Co from freshwater input

Sections 3.4.1 and 3.4.2 particularly (but also others): The sections might benefit from adding a few actual concentration ranges to the qualitative description (e.g., P16L9, 'Despite higher mesopelagic oxygen concentrations in the North Atlantic, the dissolved Co concentrations were also higher here...'; P16L14, '... have now been found to harbour high concentrations of Co'; P17L5 '...contain very elevated concentrations of Co...'). Maybe also add some concentration ranges for Fe and Mn from the literature?

P17L17 'Oxygen concentrations are also much higher in ULSW than within the Mauritianian Upwelling plume...demonstrating that low O2 in not necessarily critical to sustaining subsurface Co plumes' -> Does the higher O2 in ULSW compared to the Mauritanian OMZ go along with lower DCo concentrations? In Fig 10 you show a linear plot of LCo and DCo vs. O2 but this plot does not seem to include the stations along the western part of the transect. Does Fig. 10 include the ULSW stations you mention in this paragraph? If not, this should be mentioned in the caption.

Section 3.4.4 is a very long section, maybe subdivide to make reading easier?

P19L16 Were the shipboard aerosol samples at BATS and BATS region collected during the same cruise?

P21L26 and P22L24: You mention that Co from dust might make up a larger contribution if dissolution is slow and happens gradually from sinking particles. However, I found it very interesting that the linear DCo vs. O2 relationships in the North Atlantic and South Atlantic were comparable (Fig 10D). Does that not imply that the Co concentrations are mainly controlled by sedimentary reductive dissolution processes and water column scavenging so that the influence of dissolution of Co after atmospheric deposition is indeed negligible?

P24L8 The authors state that DCo:DMn and DFe:DMn are cosistent with diluted hydrothermal fluids. As the contribution of the hydrothermal vent to dissolved DFe and

DMn pools is much larger than for the DCo pool, do you need to subtract the surrounding 'background' DCo concentration for a comparision (also in Fig. 8)?

P24L23 A trend is observed in which Co:Mn from hydrothermal sources increases with dilution but Fe:Mn follows the opposite trend and this is disucssed in context of oxidative removal rates. However, given a much higher relative dissolved 'background' Co concentration compared to the hydrothermal input, this trend should be expected. The background Co concentration may need to be subtracted for these calculations.

P25L9 Have Al concentrations also been measured during this cruise? How strong was the MOW signal at the sampled stations?

Section 3.5 is rather complex and long to describe a variable influence of nepheloid layers. Could this section be shortened?

P26L1 Results for USGT11-06 and 11-10 are described. What about USGT11-08 where the highest suspended particle mass was measured? Maybe mention results for all three stations at the beginning of the paragraph before going into further discussion.

P27L15 'This dramatic difference where some bottom water samples along the western margin show slight enrichment while some bottom samples along the eastern margin show strong depletion...' -> I did not see this dramatic difference in Fig. 9. All profiles seem to show more or less a decrease in LCo and DCo and an increase in pCo except for station 10-09.

P28L16 The authors conclude that a similar DCo:O2 relationship in the North and South Atlantic could be related to the oxygen needs of Mn oxidation and co-oxidation. This point may deserve further explanation and a reference. By and large, I came to the understanding that DCo:O2 is governed by your previous descriptions of reductive sedimentary dissolution and scavenging after re-oxidation (biotic and abiotic).

P28L33 'implications for the ecological balance' -> Maybe you could shortly mention why this should have implications for the ecological balance or give a reference for Co

limitation here.

Section 3.7: Somewhere in this section, I was hoping for an overview of particular labile and inert Co sources or sinks, i.e. dust, sedimentary processes, uptake, etc. and maybe a discussion in context of Co oxidation states. I suggest to add this overview if it can be incorportated without adding too much text to the already long mansucript.

P30L11 'To our knowledge, this is the first report of linear relationships between labile Co and P' -> A similar linear relationship in the eastern North Atlantic has been reported previously by Baars et al, 2015.

P30L16 Maybe add references here. As mentioned above, this may not only be a question of complexation but also of oxidation of Co(II) to Co(III).

P30L16 A lower slope of LCo:P below the photic zone is reported than in the upper photic zone. I did not see this contrast in Fig. 11.

P31L3 Why are four stations chosen? Are the results with these stations representative for the whole dataset?

P31L22 What is this reference for?

P31L28 'This offset between the labile and total cobaltclines suggests that biological processes act quickly enough to complex labile Co...' -> However could this difference be simply explained by preferential uptake of labile Co acting to remove labile Co, leaving the inert, strongly complexed Co pool (see also caption in Fig. 12)?

P32L6 '...rates of uptake, complexation, diffusion,... ' -> maybe add redox reactions to this list

Figures: captions - These are rather long to summarize the main points in the text. Can these be shortened? There are too many figures. In particular there are a number of figures showing reduntant Co vs. P plots. Could Figs. 13 and 14 be moved to a SI section? Fig.6 also shows Co vs. P, maybe remove these and give a reference to Fig.

[Figure]

10 and 11. If further figures might need to be removed, maybe show Figs. 2+3 in the SI as well. I suggest to add deg W or deg N to the station numbers in all figures and text for orientation.

Fig. 1: The bathymetry contrast is not very good Fig. 7: Which depths are chosen for the surface data? Fig. 14: Give a reference for SF6 data

Technical corrections ————————— P10L22 '(Duluquais Refs)' -> Correct reference P10L13 'due it having' -> 'due to it having' P15L17 'GEOTRACES complaint' -> 'GEOTRACES compliant' P19L6 '... have displayed...' -> '... were...' P37L28 'AOU' -> has not been defined (apparent oxygen utilization) and is not mentioned in the main text P37L29 'NAZT' -> I suggest to write out the abbreviation as it is only used one more time in the ms.

---

## Referee Comment (RC2) · Anonymous Referee #2 · 9 Jan 2017

1. Does the paper address relevant scientific questions within the scope of BG?

This manuscript describes a large dataset of dissolved, labile, and particulate cobalt concentrations produced from samples collected over two US GEOTRACES section in the North Atlantic. The biogeochemical cycle of cobalt fits well within the aims and scope of the journal and this work would likely be interesting to its readership.

2. Does the paper present novel concepts, ideas, tools, or data?

The authors are at the forefront of cobalt research in the marine environment and have produced a novel dataset through their participation in the GEOTRACES program.

3. Are substantial conclusions reached?

[Figure]

The authors identify two major subsurface sources of cobalt to the study region, one each at the eastern and western margins. They also identify atmospheric deposition, riverine sources, hydrothermal activity, and nepheloid bottom layers as other factors impacting the availability of cobalt.

4. Are the scientific methods and assumptions valid and clearly outlined?

The methods are state of the art and described at length. The manuscript could be shortened if the portions directed solely at describing method improvements, e.g. Section 3.3 were spun off into a separate manuscript. I believe that such a manuscript may be in prep (pg 13, line 11).

5. Are the results sufficient to support the interpretations and conclusions?

The processes identified as possible explanations for the observed biogeochemical features are generally well justified. However, I found the text attempting to anticipate changes in cobalt availability as a function of changing oxygen concentrations to be less compelling (Section 3.6). While there is a relatively robust relationship between the concentration of dissolved cobalt and that of dissolved oxygen, I feel that the authors have failed to adequately assess the uncertainty in the calculation that leads them to predict that the inventory of cobalt could increase by as much as 20%. They make passing reference to "implications for the ecological balance within this basin" but leave the readers to guess what those implications might be. The authors conclude with a paragraph suggesting future anthropogenic cobalt pollution; do they expect this potential pollution to have a greater impact on cobalt concentrations than marine deoxygenation?

6. Is the description of experiments and calculations sufficiently complete and precise to allow their reproduction by fellow scientists (traceability of results)?

I am curious about the intercalibration efforts described in Section 2.3. One of the most important legacies of the international GEOTRACES program will be improved intercalibration among laboratories making highly precise measurements with specialized techniques. GEOTRACES and SAFe consensus samples were included in the analyses and those results were reported. However, the authors report a lack of agreement among samples that were shared with other groups as part of a GEOTRACES "crossover" station (pages 10 and 13). Perhaps the reasons for these discrepancies could be more fully explored if matters of methodology were discussed in a companion paper.

7. Do the authors give proper credit to related work and clearly indicate their own new/original contribution?

The authors give proper credit to related work including their own study in the South Atlantic as well as recent GEOTRACES-related intercomparison and intercalibration work. They also suitably reference the historical literature.

8. Does the title clearly reflect the contents of the paper?

The title is suitable.

9. Does the abstract provide a concise and complete summary?

The abstract offers a complete summary but could benefit from editing for brevity.

10. Is the overall presentation well-structured and clear?

Overall, this manuscript is well written but I encourage the authors to consider whether they would be better served by breaking the methods discussions into a companion paper or perhaps into supplementary material (see comments in #4). Without that extra discussion and Figure 4, the paper would still be impressive as it describes a large dataset covering measurements made from samples collected across the North Atlantic basin.

As it currently stands, the manuscript is quite long with a large number of figures (14) which include lengthy captions. The manuscript might also be shortened by removing

some instances of redundant text which restate material initially presented.

11. Is the language fluent and precise?

Yes.

12. Are mathematical formulas, symbols, abbreviations, and units correctly defined and used?

Yes.

13. Should any parts of the paper be clarified, reduced, combined, or eliminated?

See above.

14. Are the number and quality of references appropriate?

Yes.

15. Is the amount and quality of supplementary material appropriate?

NA.

Specific comments:

P5L1-23: This review paragraph could be eliminated without impacting the value of the manuscript.

P10L22: "Dulaquais Refs"

P10L19-26: Why mention the IDP? If the authors feel that the higher values are real, then I suggest they include those data and not discuss the intercalibration.

P11L1-9: The cruise track was described in the Methods section; there is no need to repeat.

P11L26: It appears to be that Figures 2 and 3 are reversed.

P12L17: Does the Noble and Saito manuscript focus on the analytical methods? Could

this be an appropriate companion manuscript to offload some of the methods discussion?

Figure 2 (or 3): It would be helpful to overlay the dissolved oxygen data onto the ODV section plots.

---

## Author Comment (AC2) · 10 Mar 2017

Author response: We thank Reviewer #1 for the complimentary opening statements and for taking the time to review the article in such depth. We appreciate the thoughtful comments and have responded to specific comments below:

Reviewer #2 comment: 1) The frequently used terminology 'labile Co' and 'strong Co binding ligands' mixes kinetic (labile) and thermodynamic aspects (strong ligands). For example, in P1L16, the authors conclude that strong Co binding ligands where not in excess of total Co below the euphotic zone because labile Co was measured. Based on this and other studies, as also mentioned in the introduction, it is likely that labile Co is Co(II) and inert Co is Co(III). There is a possible conversion from labile, weakly

bound Co(II) to inert, strongly bound Co(III) by oxidation. Some of the ligands that may bind Co(II) weakly may be strong ligands for Co(III) and thus it cannot be excluded that Co was remineralized as Co(II) and may, with time, be converted to inert/strongly complexed Co(III) without an apparent excess of strong Co ligands. I would suggest to add a short review of these aspects and a clarification of terminology to the speciation section in the introduction (P4L7).

Author response: Thank you, this is a thoughtful and interesting point. We can add a discussion about this and acknowledgement of these differences.

Reviewer #2 comment: 2) P6L8 Add methodology for pCo (particulate suspended Co) determination

Author response: Thank you, we will add this methodology.

Reviewer #2 comment: 3) P9L3 A detection limit of 1.8 pM for DCo is mentioned (3x standard deviation of blank) which is significantly lower than detection limits reported in earlier studies using similar methodology. Is the peak for 1.8pM actually measurable or inferred from the intercept of standard additions? A figure showing exemplary chromatograms for the blank analysis in a new Supplemental Information document might be helpful.

Author response: The detection limit is determined for each analytical dataset, and is dependent upon the stability of the electrode and how low the blank is. Yes, this is determined from the intercept following standard additions to a blank solution. For every analysis, each scan is viewed and the peak height determined. We have shown example scans in previous posters and manuscripts, including the original paper referenced in the methods section. For these reasons, we do not believe it necessary to provide example blank scans in supplemental information.

Reviewer #2 comment: 4) P10L2 'The results demonstrate that...' -> I found this sentence complicated. Do you mean that results for GEOTRACES standards are in agreement with consensus values? Author response: Yes. We will edit for clarity.

Reviewer #2 comment: 5) P10L16 'These results are in good agreement with those from the GEOTRACES intercalibration...' -> Are the results within the standard deviation of GEOTRACES consensus values? Provide a reference where these values can be found again here.

Author response: Thank you. We will add reference to the GEOTRACES consensus values again here.

Reviewer #2 comment: 6) P13L28 It is stated that the storage issue for dissolved Co is more severe in the North Atlantic than in other regions. This is very interesting and the authors suggest that it may be related to dust and colloidal loading. In general, it seems almost clear that the 'loss' of Co is because Co(II) is being oxidized to an inert Co(III) form. This can be particularly pronounced in regions with high labile Co(II) and high ligand concentrations or high colloidal concentrations that can bind Co(III). Perhaps you could mention this redox aspect again in the discussion of the storage effect.

Author response: We agree that redox processing is likely affecting preservation but think it may be more complicated, and plan to explore this in the companion methods paper. A complicating factor is that there must be other processes at play because we have also observed great stability in samples from the Ross Sea (well oxygenated waters that contain labile and complexed cobalt) where we observed labile cobalt throughout the water column, but saw no change in the labile or total concentrations when samples were re-measured, unpreserved, over a year later. If oxidation were the only factor, one would have expected a loss of the labile cobalt in those waters as well. We additionally observed no labile cobalt in arctic waters that carry high colloidal loads. Perhaps there is a catalyzing effect of colloidal and dust loading that initiates oxidation, or possibly sorbs and removed CoII without oxidation (particularly if oxidation kinetics are too slow). We are exploring these possibilities in the methods paper and have thus

omitted strong opinions from this shortened piece here, but can consider the addition of mentioning this aspect as a possible factor here.

Reviewer #2 comment: 7) P15L29 Studies are referenced showing that the OMZs in the South Atlantic and in the North Atlantic show elevated Mn and Fe. Are the elevated concentrations in the OMZs of the two regions comparable? Does Co seem to be slower to be scavenged when reaching higher O2 than Fe and Mn, similar to your discussion of the hydrothermal Co:Mn ratios (this could be added to your conclusion of a low O2 threshold for Co plume formation, P16L17)?

Author response: The concentrations are somewhat comparable but we have not worked up the ratio data to examine it. Cobalt does definitely appear to be more slowly scavenged at higher O2 than Fe and Mn, and this was a concept we discussed in the 2012 paper when comparing concentrations of Co, Mn, and Fe with distance from shore. Where the Fe and Mn concentrations dropped off quite quickly, the Cobalt concentrations persisted much further into the basin. Since we had explored this in depth in the 2012 paper, we did not repeat this discussion here.

Reviewer #2 comment: 8) P18L6 '..the relationship with salinity was similar for labile Co...' -> maybe add the fraction of labile Co from freshwater input

Author response: Do you mean to suggest that an endmember labile cobalt concentration should be calculated and proposed as the fractional contribution of labile cobalt to the total cobalt pool? If so, I'm not sure we can determine this with much confidence and it would have to be posed with caveats regarding potential labile contributions from wet deposition.

Reviewer #2 comment: 9) Sections 3.4.1 and 3.4.2 particularly (but also others): The sections might benefit from adding a few actual concentration ranges to the qualitative description (e.g., P16L9, 'Despite higher mesopelagic oxygen concentrations in the North Atlantic, the dissolved Co concentrations were also higher here...'; P16L14, '... have now been found to harbour high concentrations of Co'; P17L5 '...contain very

elevated concentrations of Co...'). Maybe also add some concentration ranges for Fe and Mn from the literature?

Author response: We did not add ranges originally as we wanted to avoid bounding these differences in a semi-quantitative fashion, when we do not mean to propose that these differences can be explicitly quantified.

Reviewer #2 comment: 10) P17L17 'Oxygen concentrations are also much higher in ULSW than within the Mauritanian Upwelling plume...demonstrating that low O2 in not necessarily critical to sustaining subsurface Co plumes' -> Does the higher O2 in ULSW compared to the Mauritanian OMZ go along with lower DCo concentrations? In Fig 10 you show a linear plot of LCo and DCo vs. O2 but this plot does not seem to include the stations along the western part of the transect. Does Fig. 10 include the ULSW stations you mention in this paragraph? If not, this should be mentioned in the caption.

Author response: These stations are not included in the figure, as stated in the figure legend. The focus of Figure 10 was to compare the major basin characteristic differences between the north and south atlantic oceans, and do not include the stations from Line W. Figure 10 is meant to focus solely on the subtropical gyres and their respective upwelling regions. This is why we do not reference Figure 10 in our discussion of ULSW. As such, we do not think it necessary to mention that Line W stations are not included in the figure caption.

Reviewer #2 comment: 11) Section 3.4.4 is a very long section, maybe subdivide to make reading easier?

Author response: Thank you. We can work on tightening this section and/or subdividing it.

Reviewer #2 comment: 12) P19L16 Were the shipboard aerosol samples at BATS and BATS region collected during the same cruise?

BGD

Author response: Yes.

Reviewer #2 comment: 13) P21L26 and P22L24: You mention that Co from dust might make up a larger contribution if dissolution is slow and happens gradually from sinking particles. However, I found it very interesting that the linear DCo vs. O2 relationships in the North Atlantic and South Atlantic were comparable (Fig 10D). Does that not imply that the Co concentrations are mainly controlled by sedimentary reductive dissolution processes and water column scavenging so that the influence of dissolution of Co after atmospheric deposition is indeed negligible?

Author response: We agree that DCo may be mainly controlled by reductive dissolution and scavenging, however; this slow input of cobalt from atmospheric deposition may have a gradual influence over time, such that the signal may lost amidst that of the other processes quickly. This offset in timescales of input makes it very difficult to state with certainty the degree of influence that atmospheric deposition may have. For these reasons, we believe there is insufficient information at this time to say that it has a negligible influence.

Reviewer #2 comment: 14) P24L8 The authors state that DCo:DMn and DFe:DMn are consistent with diluted hydrothermal fluids. As the contribution of the hydrothermal vent to dissolved DFe and DMn pools is much larger than for the DCo pool, do you need to subtract the surrounding 'background' DCo concentration for a comparison (also in Fig. 8)?

Author response: This is a good point, and in theory that would be ideal. However, in the South Atlantic, there was no discernable hydrothermal signal/feature for dCo so it was not possible to distinguish a suitable "background". In order to make the comparison between the North and South Atlantic, we need to treat the data similarly, so we could not subtract a background, but perhaps it would be worth including a caveat that the values for the North and South Atlantic Co:Mn ratios are overestimates due to the inability to distinguish the plume contribution from that of the background.

Reviewer #2 comment: 15) P24L23 A trend is observed in which Co:Mn from hydrothermal sources increases with dilution but Fe:Mn follows the opposite trend and this is discussed in context of oxidative removal rates. However, given a much higher relative dissolved 'background' Co concentration compared to the hydrothermal input, this trend should be expected. The background Co concentration may need to be subtracted for these calculations.

Author response: See response to comment 14 above. Also, thank you for highlighting this. In light of the inability to subtract the background, and the expected opposite effect that a background subtraction would have on this trend (i.e. this would lower the ratio observed in the South Atlantic more than in the North Atlantic, muting the trend), we can soften this language or remove the discussion of this trend altogether.

Reviewer #2 comment: 16) P25L9 Have Al concentrations also been measured during this cruise? How strong was the MOW signal at the sampled stations?

Author response: Yes, the paper referenced in this sentence discusses the Aluminum trends on this cruise (Measures 2015). The MOW signal was strong at these stations, as observed in the salinity profiles and Al profiles, which made the lack of a strong signal for DCo rather notable.

Reviewer #2 comment: 17) Section 3.5 is rather complex and long to describe a variable influence of nepheloid layers. Could this section be shortened?

Author response: Thank you. We can make an effort to condense this section.

Reviewer #2 comment: 18) P26L1 Results for USGT11-06 and 11-10 are described. What about USGT11-08 where the highest suspended particle mass was measured? Maybe mention results for all three stations at the beginning of the paragraph before going into further discussion.

Author response: Thank you. We will take this into consideration and include a description of the observed SPM at USGT11-08.
**BGD**

Reviewer #2 comment: 19) P27L15 'This dramatic difference where some bottom water samples along the western margin show slight enrichment while some bottom samples along the eastern margin show strong depletion...' -> I did not see this dramatic difference in Fig. 9. All profiles seem to show more or less a decrease in LCo and DCo and an increase in pCo except for station 10-09.

Author response: We are focused on the very deepest 1-3 samples for each station. On the eastern margin, there is a decrease in labile cobalt for all three stations to concentrations in the single digits, where there is much less of a decrease in the western margin, with labile concentrations hovering around 20pM. The slight increase in labile and total cobalt at USGT11-10 is a dramatic contrast to the strong depletion of total and labile cobalt at USGT10-09. We will clarify this contrast so that it focuses more precisely on the two stations that differ dramatically.

Reviewer #2 comment: 20) P28L16 The authors conclude that a similar DCo:O2 relationship in the North and South Atlantic could be related to the oxygen needs of Mn oxidation and co-oxidation. This point may deserve further explanation and a reference. By and large, I came to the understanding that DCo:O2 is governed by your previous descriptions of reductive sedimentary dissolution and scavenging after re-oxidation (biotic and abiotic).

Author response: Thank you. Yes, we believe the signal is governed by both of these processes (reductive dissolution + scavenging/oxidation). The suggestion here was that the relationship might be maintained by the oxygen demands of Mn oxidation and co-oxidation (i.e. that while these processes all govern the observed trend, the concentration of oxygen might influence the activity of manganese oxidizing bacteria which we think are responsible for a substantial portion of the cobalt removal (scavenging via biotic co-oxidation by mn-oxidizing bacteria). We can add a sentence or rewrite the current one for clarity and add a reference.

Reviewer #2 comment: 21) P28L33 'implications for the ecological balance' -> Maybe

you could shortly mention why this should have implications for the ecological balance or give a reference for Co limitation here.

Author response: Thank you. We will address this concern.

Reviewer #2 comment: 22) Section 3.7: Somewhere in this section, I was hoping for an overview of particular labile and inert Co sources or sinks, i.e. dust, sedimentary processes, uptake, etc. and maybe a discussion in context of Co oxidation states. I suggest to add this overview if it can be incorporated without adding too much text to the already long manuscript.

Author response: Thank you for the suggestion. We will consider if/how this could be incorporated.

Reviewer #2 comment: 23) P30L11 'To our knowledge, this is the first report of linear relationships between labile Co and P' -> A similar linear relationship in the eastern North Atlantic has been reported previously by Baars et al, 2015.

Author response: Thank you for bringing this to our attention. We will review the article and be sure to add the citation.

Reviewer #2 comment: 24) P30L16 Maybe add references here. As mentioned above, this may not only be a question of complexation but also of oxidation of Co(II) to Co(III).

Author response: Thank you. we will consider adding references.

Reviewer #2 comment: 25) P30L16 A lower slope of LCo:P below the photic zone is reported than in the upper photic zone. I did not see this contrast in Fig. 11.

Author response: We will try to clarify the text. The contrast is between the slopes defined by the linear regression of the dCo:P below the photic zone (black circles for USGT scatter plots, 41-67 umol/mol) and the slopes defined by the linear regression of LCo:P (black triangles for USGT scatter plots, slopes of 19-28). We were trying to highlight the large difference between a slope range of 41-67 and 19-28.

Reviewer #2 comment: 26) P31L3 Why are four stations chosen? Are the results with these stations representative for the whole dataset?

Author response: These stations were meant to illustrate how upwelling and oligotrophic waters can have offsets in the response expected trends in Co:P relationships for labile and total cobalt. They are not meant to be representative of the whole dataset, but rather, to best illustrate the range observed here from an oligotrophic region in which the labile and total cobaltclines are offset, and in an upwelling region where they are coincident, and the variability in between where the mixed layer depth and chlorophyll maximum lend clues to the controls on total and labile cobalt.

Reviewer #2 comment: 27) P31L22 What is this reference for?

Author response: Thank you for noticing this. This reference addresses a similar phenomenon but is not well placed. We will clarify and move it accordingly.

Reviewer #2 comment: 28) P31L28 'This offset between the labile and total cobaltclines suggests that biological processes act quickly enough to complex labile Co...' -> However could this difference be simply explained by preferential uptake of labile Co acting to remove labile Co, leaving the inert, strongly complexed Co pool (see also caption in Fig. 12)?

Author response: Yes, that could be a piece of it. Both labile and complexed cobalt are bioavailable, but it has been demonstrated that some organisms only have access to the labile fraction. When we say that biological processes act quickly enough to complex labile cobalt, this process may occur via uptake of the labile fraction and subsequent complexation or utilization. Complexation of labile cobalt by biota that can utilize complexed cobalt could give those organisms a competitive edge and preferential access to the total cobalt pool.

Reviewer #2 comment: 29) P32L6 '...rates of uptake, complexation, diffusion,... ' -> maybe add redox reactions to this list

Author response: Thank you, we will consider adding

Reviewer #2 comment: 30) Figures: captions - These are rather long to summarize the main points in the text. Can these be shortened?

Author response: We will shorten the captions.

Reviewer #2 comment: There are too many figures. In particular there are a number of figures showing reduntant Co vs. P plots. Could Figs. 13 and 14 be moved to a SI section? Fig.6 also shows Co vs. P, maybe remove these and give a reference to Fig. 10 and 11.

Author response: We respectfully disagree regarding Co:P redundancy. Each plot specifically addresses a different trend and aspect of the data: Fig6: lack of correlation between Co:P for the Line W data relative to the gyre. (Line W data are excluded from the other Co:P plots) Fig10: global trends in Co:P (C) and the offset between the N and S Atlantic, paired with the respective N:P data (B). Fig 11: differences in slope between (1 above and below the photic zone, (2 gyres and upwellings, and 3) labile and total cobalt. We do not think that any of the above points can be made successfully without a figure, and the figures cannot be condensed as they would not clearly illustrate these very different points. Perhaps Fig13 could be moved to the SI if necessary, though it would require the reader to access the SI to fully understand the paper.

Reviewer #2 comment: If further figures might need to be removed, maybe show Figs. 2+3 in the SI as well.

Author response: Figure 2 should not go in the supplemental material as it articulates the overall oceanic distribution and the dimensions and parameters for the figure were specified at GEOTRACES meetings for AG03 in order to assure comparability among manuscripts. Perhaps Figure 3 could be condensed for the manuscript, showing selected profiles, but displayed in full in the SI.

Reviewer #2 comment: I suggest to add deg W or deg N to the station numbers in all

figures and text for orientation.

Author response: The figures are already rather text heavy. Adding ° W and ° N would add considerable clutter.

Reviewer #2 comment: Fig. 1: The bathymetry contrast is not very good Author response: Due to the bathymetry being included in Figure 2, we do not think this is a significant problem, but thank you for pointing it out. Reviewer #2 comment: Fig. 7: Which depths are chosen for the surface data? Author response: These samples are the surface fish samples. This is implied in the figure caption, but we will clarify further. Reviewer #2 comment: Fig. 14: Give a reference for SF6 data Author response: Thank you, we will address Reviewer #2 comment: P10L22 '(Duluquais Refs)' -> Correct reference Author response: Thank you, we will correct this. Reviewer #2 comment: P10L13 'due it having' -> 'due to it having' Author response: Thank you, we will correct this. Reviewer #2 comment: P15L17 'GEOTRACES complaint' ->'GEOTRACES compliant' Author response: Thank you, we will correct this. Reviewer #2 comment: P19L6 '... have displayed...' -> '... were...' Author response: Thank you, we will correct this. Reviewer #2 comment: P37L28 'AOU' -> has not been defined (apparent oxygen utilization) and is not mentioned in the main text Author response: Thank you. We will address this. Reviewer #2 comment: P37L29 'NAZT' -> I suggest to write out the abbreviation as it is only used one more time in the ms. Author response: Yes, we will fix this.

---

## Author Response (AR1)

**Point-by-point addressed comments from reviewers:**

Reviewer #1 comment: 1) The frequently used terminology 'labile Co' and 'strong Co binding ligands' mixes kinetic (labile) and thermodynamic aspects (strong ligands). For example, in P1L16, the authors conclude that strong Co binding ligands where not in excess of total Co below the euphotic zone because labile Co was measured. Based on this and other studies, as also mentioned in the introduction, it is likely that labile Co is Co(II) and inert Co is Co(III). There is a possible conversion from labile, weakly bound Co(II) to inert, strongly bound Co(III) by oxidation. Some of the ligands that may bind Co(II) weakly may be strong ligands for Co(III) and thus it cannot be excluded that Co was remineralized as Co(II) and may, with time, be converted to inert/strongly complexed Co(III) without an apparent excess of strong Co ligands. I would suggest to add a short review of these aspects and a clarification of terminology to the speciation section in the introduction (P4L7).

Author response: We have added a clarifying discussion regarding the definition of different ligand classes, our observations in the field, previous observations of CoII and CoIII, and how we suspect that redox plays a part in the ocean cycling of cobalt.

Reviewer #1 comment: 2) P6L8 Add methodology for pCo (particulate suspended Co) determination

Author response: We have added this methodology.

Reviewer #1 comment: 3) P9L3 A detection limit of 1.8 pM for DCo is mentioned (3x standard deviation of blank) which is significantly lower than detection limits reported in earlier studies using similar methodology. Is the peak for 1.8pM actually measurable or inferred from the intercept of standard additions? A figure showing exemplary chromatograms for the blank analysis in a new Supplemental Information document might be helpful.

Author response: The detection limit is determined for each analytical dataset, and is dependent upon the stability of the electrode and how low the blank is. Yes, this is determined from the intercept following standard additions to a blank solution. For every analysis, each scan is viewed and the peak height determined. We have shown example scans in previous posters and manuscripts, including the original paper referenced in the methods section. For these reasons, we did not provide example blank scans in supplemental information.

Reviewer #1 comment: 4) P10L2 'The results demonstrate that...' -> I found this sentence complicated. Do you mean that results for GEOTRACES standards are in agreement with consensus values?

Author response: We added clarifying text.

Reviewer #1 comment: 5) P10L16 'These results are in good agreement with those from the GEOTRACES intercalibration...'
-> Are the results within the standard deviation of GEOTRACES consensus values? Provide a reference where these values can be found again here.

Author response: We have added reference to the GEOTRACES consensus values again here.

Reviewer #1 comment: 6) P13L28 It is stated that the storage issue for dissolved Co is more severe in the North Atlantic than in other regions. This is very interesting and the authors suggest that it may be related to dust and colloidal loading. In general, it seems almost clear that the 'loss' of Co is because Co(II) is being oxidized to an inert Co(III) form. This can be particularly pronounced in regions with high labile Co(II) and high ligand concentrations or high colloidal concentrations that can bind Co(III). Perhaps you could mention this redox aspect again in the discussion of the storage effect.

Author response: We agree that redox processing is likely affecting preservation but think it may be more complicated, and plan to explore this in the companion methods paper. We have added an explicit mention of the oxidation of CoII to CoIII in high colloidal regions and reiterated this.

Reviewer #1 comment: 7) P15L29 Studies are referenced showing that the OMZs in the South Atlantic and in the North Atlantic show elevated Mn and Fe. Are the elevated concentrations in the OMZs of the two regions comparable? Does Co seem to be slower to be scavenged when reaching higher O2 than Fe and Mn, similar to your discussion of the hydrothermal Co:Mn ratios (this could be added to your conclusion of a low O2 threshold for Co plume formation, P16L17)?

Author response: The concentrations are somewhat comparable but we have not worked up the ratio data to examine it. Cobalt does definitely appear to be more slowly scavenged at higher O2 than Fe and Mn, and this was a concept we discussed in the 2012 paper when comparing concentrations of Co, Mn, and Fe with distance from shore. Where the Fe and Mn concentrations dropped off quite quickly, the Cobalt concentrations persisted much further into the basin. Since we had explored this in depth in the 2012 paper, we did not repeat this discussion here, but have now added mention and reference to this discussion for clarity and completeness.

Reviewer #1 comment: 8) P18L6 '..the relationship with salinity was similar for labile Co...' -> maybe add the fraction of labile Co from freshwater input

Author response: We do not thing we can determine this with much confidence and it would have to be posed with caveats regarding potential labile contributions from wet deposition.

Reviewer #1 comment: 9) Sections 3.4.1 and 3.4.2 particularly (but also others): The sections might benefit from adding a few actual concentration ranges to the qualitative description (e.g., P16L9, 'Despite higher mesopelagic oxygen concentrations in the North Atlantic, the dissolved Co concentrations were also higher here...'; P16L14, '... have now been found to harbour high concentrations of Co'; P17L5 '...contain very elevated concentrations of Co...'). Maybe also add some concentration ranges for Fe and Mn from the literature?

Author response: We did not add ranges originally as we wanted to avoid bounding these differences in a semi-quantitative fashion, when we do not mean to propose that these differences can be explicitly quantified.

Reviewer #1 comment: 10) P17L17 'Oxygen concentrations are also much higher in ULSW than within the Mauritanian Upwelling plume...demonstrating that low O2 in not necessarily critical to sustaining subsurface Co plumes' -> Does the higher O2 in ULSW compared to the Mauritanian OMZ go along with lower DCo concentrations? In Fig 10 you show a linear plot of LCo and DCo vs. O2 but this plot does not seem to include the stations along the western part of the transect. Does Fig. 10 include the ULSW stations you mention in this paragraph? If not, this should be mentioned in the caption.

Author response: These stations are not included in the figure, as stated in the figure legend. The focus of Figure 10 was to compare the major basin characteristic differences between the north and south atlantic oceans, and do not include the stations from Line W. Figure 10 is meant to focus solely on the subtropical gyres and their respective upwelling regions. This is why we do not reference Figure 10 in our discussion of ULSW. As such, we do not think it necessary to mention that Line W stations are not included in the figure caption.

Reviewer #1 comment: 11) Section 3.4.4 is a very long section, maybe subdivide to make reading easier?

Author response: Thank you. We have tightened and sub-divided this section.

Reviewer #1 comment: 12) P19L16 Were the shipboard aerosol samples at BATS and BATS region collected during the same cruise?

Author response: Yes.

Reviewer #1 comment: 13) P21L26 and P22L24: You mention that Co from dust might make up a larger contribution if dissolution is slow and happens gradually from sinking particles. However, I found it very interesting that the linear DCo vs. O2 relationships in the North Atlantic and South Atlantic were comparable (Fig 10D). Does that not imply that the Co concentrations are mainly controlled by sedimentary reductive dissolution processes and water column scavenging so that the influence of dissolution of Co after atmospheric deposition is indeed negligible?

Author response: We agree that DCo may be mainly *controlled* by reductive dissolution and scavenging, however; this slow input of cobalt from atmospheric deposition may have a gradual influence over time, such that the signal may lost amidst that of the other processes quickly. This offset in timescales of input makes it very difficult to state with certainty the degree of influence that atmospheric deposition may have. For these reasons, we believe there is insufficient information at this time to say that it has a negligible influence.

Reviewer #1 comment: 14) P24L8 The authors state that DCo:DMn and DFe:DMn are consistent with diluted hydrothermal fluids. As the contribution of the hydrothermal vent to dissolved DFe and DMn pools is much larger than for the DCo pool, do you need to subtract the surrounding 'background' DCo concentration for a comparison (also in Fig. 8)?

Author response: This is a good point, and in theory that would be ideal. However, in the South Atlantic, there was no discernable hydrothermal signal/feature for dCo so it was not possible to distinguish a suitable "background". In order to make the comparison between the North and South Atlantic, we need to treat the data similarly, so we could not subtract a background.

Reviewer #1 comment: 15) P24L23 A trend is observed in which Co:Mn from hydrothermal sources increases with dilution but Fe:Mn follows the opposite trend and this is discussed in context of oxidative removal rates. However, given a much higher relative dissolved 'background' Co concentration compared to the hydrothermal input, this trend should be expected. The background Co concentration may need to be subtracted for these calculations.

Author response: See response to comment 14 above. Also, thank you for highlighting this. In light of the inability to subtract the background, and the expected opposite effect that a background subtraction would have on this trend (*i.e.* this would lower the ratio observed in the South Atlantic more than in the North Atlantic, muting the trend), we have removed the discussion of this trend.

Reviewer #1 comment: 16) P25L9 Have Al concentrations also been measured during this cruise? How strong was the MOW signal at the sampled stations?

Author response: Yes, the paper referenced in this sentence discusses the Aluminum trends on this cruise (Measures 2015). The MOW signal was strong at these stations, as observed in the salinity profiles and Al profiles, which made the lack of a strong signal for DCo rather notable.

Reviewer #1 comment: 17) Section 3.5 is rather complex and long to describe a variable influence of nepheloid layers. Could this section be shortened?

Author response: Thank you. We have condensed this section.

Reviewer #1 comment: 18) P26L1 Results for USGT11-06 and 11-10 are described. What about USGT11-08 where the highest suspended particle mass was measured? Maybe mention results for all three stations at the beginning of the paragraph before going into further discussion.

Author response: Thank you. We considered and addressed this in our effort to condense, shorten, and clarify the main points of this section.

Reviewer #1 comment: 19) P27L15 'This dramatic difference where some bottom water samples along the western margin show slight enrichment while some bottom samples along the eastern margin show strong depletion...' -> I did not see this dramatic difference in Fig. 9. All profiles seem to show more or less a decrease in LCo and DCo and an increase in pCo except for station 10-09.

Author response: We are focused on the very deepest 1-3 samples for each station. On the eastern margin, there is a decrease in labile cobalt for all three stations to concentrations in the single digits, where there is much less of a decrease in the western margin, with labile concentrations hovering around 20pM. The slight increase in labile and total cobalt at USGT11-10 is a dramatic contrast to the strong depletion of total and labile cobalt at USGT10-09. We have clarified this contrast so that it focuses more precisely on the two stations that differ dramatically.

Reviewer #1 comment: 20) P28L16 The authors conclude that a similar DCo:O2 relationship in the North and South Atlantic could be related to the oxygen needs of Mn oxidation and co-oxidation. This point may deserve further explanation and a reference. By and large, I came to the understanding that DCo:O2 is governed by your previous descriptions of reductive sedimentary dissolution and scavenging after re-oxidation (biotic and abiotic).

Author response: Yes, we believe the signal is governed by both of these processes (reductive dissolution + scavenging/oxidation). The suggestion here was that the relationship might be maintained by the oxygen demands of Mn oxidation and co-oxidation (*i.e.* that while these processes all govern the observed trend, the concentration of oxygen might influence the activity of manganese oxidizing bacteria which we think are responsible for a substantial portion of the cobalt removal (scavenging *via* biotic co-oxidation by mn-oxidizing bacteria). We edited this sentence for clarity.

Reviewer #1 comment: 21) P28L33 'implications for the ecological balance' -> Maybe you could shortly mention why this should have implications for the ecological balance or give a reference for Co limitation here.

Author response: Thank you. We added a sentence and reference.

Reviewer #1 comment: 22) Section 3.7: Somewhere in this section, I was hoping for an overview of particular labile and inert Co sources or sinks, i.e. dust, sedimentary processes, uptake, etc. and maybe a discussion in context of Co oxidation states. I suggest to add this overview if it can be incorporated without adding too much text to the already long manuscript.

Author response: Thank you for the suggestion. We added a few sentences to address what is known and what remains to be understood regarding the different contributions to the chemical speciation of dissolved cobalt from various sources and sinks.

Reviewer #1 comment: 23) P30L11 'To our knowledge, this is the first report of linear relationships between labile Co and P' -> A similar linear relationship in the eastern North Atlantic has been reported previously by Baars et al, 2015.

Author response: Thank you for bringing this to our attention. We have removed this claim.

Reviewer #1 comment: 24) P30L16 Maybe add references here. As mentioned above, this may not only be a question of complexation but also of oxidation of Co(II) to Co(III).

Author response: Thank you. We respectfully disagree, though we believe we have addressed the reviewers concerns more fully in our response and added clarity in the text regarding comments 1 and 6. We do not believe that the bioavailability of cobalt if affected much by cobalt (II) ligands unless Nickel, which tends to bind more strongly to these ligands (Saito *et al*. 2005), were completely saturated. this would be the only incidence under which excess ligands would be available for binding of Co(II).

Reviewer #1 comment: 25) P30L16 A lower slope of LCo:P below the photic zone is reported than in the upper photic zone. I did not see this contrast in Fig. 11.

Author response: We have clarified this in the text. The contrast is between the slopes defined by the linear regression of the dCo:P below the photic zone (black circles for USGT scatter plots, 41-67 umol/mol) and the slopes defined by the linear regression of LCo:P (black triangles for USGT scatter plots, slopes of 19-28).

Reviewer #1 comment: 26) P31L3 Why are four stations chosen? Are the results with these stations representative for the whole dataset?

Author response: These stations were meant to illustrate how upwelling and oligotrophic waters can have offsets in the response expected trends in Co:P relationships for labile and total cobalt. They are not meant to be representative of the whole dataset, but rather, to best illustrate the range observed here from an oligotrophic region in which the labile and total cobaltclines are offset, and in an upwelling region where they are coincident, and the variability in between where the mixed layer depth and chlorophyll maximum lend clues to the controls on total and labile cobalt.

Reviewer #1 comment: 27) P31L22 What is this reference for?

Author response: Thank you for noticing this. This reference addresses a similar phenomenon but was poorly placed and has now been removed.

Reviewer #1 comment: 28) P31L28 'This offset between the labile and total cobaltclines suggests that biological processes act quickly enough to complex labile Co...' -> However could this difference be simply explained by preferential uptake of labile Co acting to remove labile Co, leaving the inert, strongly complexed Co pool (see also caption in Fig. 12)?

Author response: Yes, that could be a piece of it. Both labile and complexed cobalt are bioavailable, but it has been demonstrated that some organisms only have access to the labile fraction. When we say that biological processes act quickly enough to complex labile cobalt, this process may occur *via* uptake of the labile fraction and subsequent complexation or utilization. Complexation of labile cobalt by biota that can utilize complexed cobalt could give those organisms a competitive edge and preferential access to the total cobalt pool.

Reviewer #1 comment: 29) P32L6 '...rates of uptake, complexation, diffusion,... ' -> maybe add redox reactions to this list

Author response: We have added this.

Reviewer #1 comment: 30) Figures: captions - These are rather long to summarize the main points in the text. Can these be shortened?
Author response: We have edited the captions for brevity.

Reviewer #1 comment: There are too many figures. In particular there are a number of figures showing reduntant Co vs. P plots. Could Figs. 13 and 14 be moved to a SI section? Fig.6 also shows Co vs. P, maybe remove these and give a reference to Fig. 10 and 11.

Author response: We respectfully disagree regarding Co:P redundancy. Each plot specifically addresses a different trend and aspect of the data:
Fig6: lack of correlation between Co:P for the Line W data relative to the gyre. (Line W data are excluded from the other Co:P plots)
Fig10: global trends in Co:P (C) and the offset between the N and S Atlantic, paired with the respective N:P data (B).
Fig 11: differences in slope between (1 above and below the photic zone, (2 gyres and upwellings, and 3) labile and total cobalt.
We do not think that any of the above points can be made successfully without a figure, and the figures cannot be condensed as they would not clearly illustrate these very different points. Perhaps Fig13 could be moved to an SI if requested by the editor, though it would require the reader to access the SI to fully understand the paper. For the sake of completeness though, we would prefer to keep the figures as they are.

Reviewer #1 comment: If further figures might need to be removed, maybe show Figs. 2+3 in the SI as well.

Author response: Figure 2 should not go in a supplemental section as it articulates the overall oceanic distribution and the dimensions and parameters for the figure were specified at GEOTRACES meetings for AG03 in order to assure comparability among manuscripts. Again, for the sake of completeness and due to the large dataset, we would prefer to keep the figures as they are. One of the great benefits of this journal is the online format and this was a motivating factor in our decision to submit here as we could take advantage of the large format to present the entire dataset. If the editor instructs us to do so, we will make an effort to move some of the profiles to a supplemental section.

Reviewer #1 comment: I suggest to add deg W or deg N to the station numbers in all figures and text for orientation.

Author response: The figures are already rather text heavy. Adding $^\circ$ W and $^\circ$ N would add considerable clutter.

Reviewer #1 comment: Fig. 1: The bathymetry contrast is not very good

Author response: Due to the bathymetry being included in Figure 2, we do not think this is a significant problem, but thank you for pointing it out.

Reviewer #1 comment: Fig. 7: Which depths are chosen for the surface data?

Author response: We have now clarified this in the caption.

Reviewer #1 comment: Fig. 14: Give a reference for SF6 data

Author response: Done.

Reviewer #1 comment: P10L22 '(Duluquais Refs)' -> Correct reference

Author response: Thank you, we have corrected this.

Reviewer #1 comment: P10L13 'due it having' -> 'due to it having'

Author response: Done.

Reviewer #1 comment: P15L17 'GEOTRACES complaint' ->'GEOTRACES compliant'

Author response: Done.

Reviewer #1 comment: P19L6 '... have displayed...' -> '... were...'

Author response: Done.

Reviewer #1 comment: P37L28 'AOU' -> has not been defined (apparent oxygen utilization) and is not mentioned in the main text

Author response: Done.

Reviewer #1 comment: P37L29 'NAZT' -> I suggest to write out the abbreviation as it is only used one more time in the ms.

Author response: Done.

Reviewer #2 comment: The manuscript could be shortened if the portions directed solely at describing method improvements, e.g. Section 3.3 were spun off into a separate manuscript. I believe that such a manuscript may be in prep (pg 13, line 11).

Author response: Referee #2 suggests that the information regarding intercalibration and preservation be presented in a separate manuscript. We respectfully believe that this section is important to include here due to the slight departure from our previous approaches, the results showing successful preservation, and to support the extent to which these data are compared to previous work in the South Atlantic that employed the original methods. This is important for the credibility of the comparison discussion. Additionally, the trace metal community has been asking about this data for several years and we feel it is overdue to be published. The first author on this paper has moved to a different field of research and as a result, it may take a much longer time for us to coordinate to get the full methods/preservation work published. The figure presented here is just one of several pieces of the methods work, but we feel that it represents an important piece related to GEOTRACES intercalibration that needs to be recognized sooner rather than later.

Reviewer #2 comment: I found the text attempting to anticipate changes in cobalt availability as a function of changing oxygen concentrations to be less compelling (Section 3.6). While there is a relatively robust relationship between the concentration of dissolved cobalt and that of dissolved oxygen, I feel that the author have failed to adequately assess the uncertainty in the calculation that leads them to predict that the inventory of cobalt could increase by as much as 20%. They make passing reference to "implications for the ecological balance within this basin" but leave the readers to guess what those implications might be.

Author response: Yes, rigorous statistics were not performed to allow for an assessment of the uncertainty; however, our conclusions are couched in soft language, suggesting only that the results of our calculations "imply a need to consider the influence of changing oceanic oxygen on the biogeochemistries of metals and their influence on marine ecology." (pg 29 line 3). We have previously applied this type of assessment to data in a prior publication (Noble *et al*. 2012) and do not believe this is an overreach.

Reviewer #2 comment: The authors conclude with a paragraph suggesting future anthropogenic cobalt pollution; do they expect this potential pollution to have a greater impact on cobalt concentrations than marine deoxygenation?

Author response: As noted by the reviewer, there is considerable uncertainty regarding the magnitude of the proposed potential impacts on cobalt distribution (marine deoxygenation *vs*. anthropogenic pollution) and we feel that opining on this magnitude would be an overreach.

Reviewer #2 comment: I am curious about the intercalibration efforts described in Section 2.3. One of the most important legacies of the international GEOTRACES program will be improved inter-calibration among laboratories making highly precise measurements with specialized techniques. GEOTRACES and SAFe consensus samples were included in the analyses and those results were reported. However, the authors report a lack of agreement among samples that were shared with other groups as part of a GEOTRACES "crossover" station (pages 10 and 13). Perhaps the reasons for these discrepancies could be more fully explored if matters of methodology were discussed in a companion paper.

Author response: Thank you. We also believe that these discrepancies are related to the preservation methodologies and thus cannot be fully discussed here without a full discussion of the preservation methodology. As such, these issues will be covered in the methods/preservation paper in preparation.

Reviewer #2 comment: The abstract offers a complete summary but could benefit from editing for brevity.

Author response: Thank you, we have edited for brevity.

Reviewer #2 comment: Overall, this manuscript is well written but I encourage the authors to consider whether
they would be better served by breaking the methods discussions into a companion
paper or perhaps into supplementary material (see comments in #4). Without that
extra discussion and Figure 4, the paper would still be impressive as it describes a
large dataset covering measurements made from samples collected across the North
Atlantic basin. As it currently stands, the manuscript is quite long with a large number of figures (14)
which include lengthy captions. The manuscript might also be shortened by removing some instances of
redundant text which restate material initially presented.

Author response: Thank you. We have shortened the figure captions.  Should the editor require it, we will add a supplementary material section, however, for the sake of completeness and given the online format of this journal, we would prefer to keep the information all present in the manuscript as it is a large dataset and the figures are meant to illustrate the discussion points.

Reviewer #2 comment: P5L1-23: This review paragraph could be eliminated without impacting the value of the manuscript.

Author response: We thought it valuable to provide a review for those readers without a background in cobalt biogeochemistry.  However; we can condense this paragraph if required by the editor.

Reviewer #2 comment: P10L22: "Dulaquais Refs"

Author response: Thank you. We have fixed this.

Reviewer #2 comment: P10L19-26: Why mention the IDP? If the authors feel that the higher values are real, then I suggest they include those data and not discuss the intercalibration.

Author response: Because cobalt had not previously been considered one of the GEOTRACES key constituents, we believe intercalibration efforts to be important to acknowledge in the effort to have cobalt included among the key trace elements and isotopes. Additionally, GEOTRACES papers typically include this intercalibration data, it is not much to add space-wise, and we therefore believe it warranted to include the intercalibration data here.

Reviewer #2 comment: P11L1-9: The cruise track was described in the Methods section; there is no need to repeat.

Author response: Thank you. We have significantly shortened this.

Reviewer #2 comment: P11L26: It appears to be that Figures 2 and 3 are reversed.

Author response: Thank you, we have fixed this reference

Reviewer #2 comment: P12L17: Does the Noble and Saito manuscript focus on the analytical methods? Could this be an appropriate companion manuscript to offload some of the methods discussion?

Author response: Again, please see above response regarding companion manuscript. Yes this manuscript will focus on analytical methods, but also addresses other aspects of the results therein.

Reviewer #2 comment: Figure 2 (or 3): It would be helpful to overlay the dissolved oxygen data onto the ODV section plots.

Author response: We agree that the concept of an oxygen overlay would be helpful perhaps in a separate figure, but this figure (3) is already quite busy and an overlay of dissolved oxygen would be too distracting and would make the figure too cluttered.

[revised manuscript text omitted]

Contours – Silicate [μM]

[Figure]

Distance [km]

**Figure 6**

[Figure]

**Figure 7**

[Figure]

**Figure 8**

[Figure]

**Figure 9**

[Figure]

**Figure 10**

[Figure]

**Figure 11**

[Figure]

**Figure 12**

[Figure]

**Figure 13**

[Figure]

**Figure 14**

[Figure]